# Premature skewing of T cell receptor clonality and delayed memory expansion in HIV-exposed infants

Sonwabile Dzanibe[1,2], Aaron J. Wilk [3], Susan Canny[3,4], Thanmayi Ranganath [3], Berenice Alinde[5], Florian Rubelt[6,7], Huang Huang[7], Mark M. Davis [6,7,8], Susan P. Holmes[9], Heather B. Jaspan [1,2,10] ✉, Catherine A. Blish [3,7,11] ✉ & Clive M. Gray [2,5] ✉

While preventing vertical HIV transmission has been very successful, HIV-exposed uninfected infants (iHEU) experience an elevated risk to infections compared to HIV-unexposed and uninfected infants (iHUU). Here we present a longitudinal multimodal analysis of infant immune ontogeny that highlights the impact of HIV/ARV exposure. Using mass cytometry, we show alterations in T cell memory differentiation between iHEU and iHUU being significant from week 15 of life. The altered memory T cell differentiation in iHEU was preceded by lower TCR Vβ clonotypic diversity and linked to TCR clonal depletion within the naïve T cell compartment. Compared to iHUU, iHEU had elevated CD56$^{lo}$CD16$^{lo}$Perforin$^+$CD38$^+$CD45RA$^+$FcεRIγ$^+$ NK cells at 1 month postpartum and whose abundance pre-vaccination were predictive of vaccine-induced pertussis and rotavirus antibody responses post 3 months of life. Collectively, HIV/ARV exposure disrupted the trajectory of innate and adaptive immunity from birth which may underlie relative vulnerability to infections in iHEU.

Early life, especially in Africa, is often plagued by high infectious morbidity with infectious diseases accounting for over two million deaths of which 45% occur during the neonatal period[1]. This high mortality rate is most likely linked to the period during which the immune system adapts to extrauterine life. During this transition window, several factors such as maternal morbidity, microbial exposure, and vaccination significantly modulate infant immunity and consequently influences health and disease outcomes[2–5]. Investigating immune ontogeny and factors that impact immune trajectory during

infancy can provide important insights into understanding how better to combat these early life immune stressors.

Current dogma is that the in-utero environment is sterile and that newborn infants have limited antigen exposure prior to birth, with potentially protective T cells being predominantly naïve and having little T cell receptor (TCR) engagement[6,7]. This lack of pre-existing adaptive cellular memory early in life is likely to increase the vulnerability of infants to infectious agents and disease, especially if there is an absence of adequate breast-feeding to provide passive maternal

[1]Division of Immunology, Department of Pathology, University of Cape Town, Cape Town, South Africa. [2]Institute of Infectious Disease and Molecular Medicine, University of Cape Town, Cape Town, South Africa. [3]Department of Medicine, School of Medicine, Stanford University, Stanford, CA, USA. [4]Division of Rheumatology, Department of Pediatrics, Seattle Children's Hospital, Seattle, WA, USA. [5]Division of Immunology, Department of Biomedical Sciences, Biomedical Research Institute, Stellenbosch University, Cape Town, South Africa. [6]Department of Microbiology and Immunology, Stanford University School of Medicine, Stanford, CA, USA. [7]Institute for Immunity, Transplantation and Infection, Stanford University School of Medicine, Stanford, CA, USA. [8]Howard Hughes Medical Institute, School of Medicine, Stanford University, Stanford, CA, USA. [9]Department of Statistics, Stanford University, Stanford, CA, USA. [10]Seattle Children's Research Institute and Department of Paediatrics and Global Health, University of Washington, Seattle, WA, USA. [11]Chan Zuckerberg Biohub, San Francisco, CA, USA. ✉e-mail: hbjaspan@gmail.com; cblish@stanford.edu; cgray@sun.ac.za

antibody immunity[3,8,9]. Since the diversity and composition of T cell immunity is shaped in large part by antigen exposure[10], examining changes in the TCR repertoire during infancy could provide insight into immune development and how disruption of immunity may influence susceptibility to infectious diseases.

The advent of universal treatment with antiretroviral (ARV) drugs to both mother and infant has successfully minimised vertical HIV transmission, but also introduces the possible adverse effect of HIV and ARV exposure on neonatal development *in utero* and in the post-natal period. The intertwined nature of HIV and ARV exposure makes it difficult to unravel, but it is known that HIV/ARV-exposed uninfected infants (iHEU) have higher risk of infectious disease-related morbidity and mortality compared to HIV-unexposed uninfected infants (iHUU) of a similar age, suggesting disruptions to their immune maturation[11–16]. Many studies investigating immunological disparities between iHEU and iHUU are limited to cross-sectional analysis with few studies investigating longitudinal changes and thus there is a lack of evidence on how T cells mature early in life in the context of HIV/ARV exposure. We have previously shown that maternal HIV/ARV exposure alters the dynamics of the T regulatory (Treg) to Th17 cell ratio resulting in a Th17/Treg imbalance associated with gut damage[17]. In this paper, we extend this analysis to investigate the impact of HIV/ARV exposure on T cell clonality, memory and NK cell maturation differences between iHEU and iHUU.

Since early life is also marked by regulated adaptive immunity, neonatal immune defence is heavily reliant on innate immune cells such as NK cells to rapidly eliminate infections. Compared to adults, however, neonatal NK cells display functional defects such as reduced cytolytic activity including antibody mediated cell cytotoxicity[18,19], decreased expression of adhesion molecules[20] and lower secretion of TNF-α and IFN-γ[21]. Earlier studies have further demonstrated that compared to iHUU, iHEU have a lower proportion of NK cells measured at birth and at six months and having reduced IFN-γ secretion and perforin expression[22,23]. In addition, the function of NK cells is influenced by the combinatorial signalling through a diverse array of activating and inhibitory receptors expressed on the cell surface[24]. The composition of these receptors is driven by past viral exposure[21,25,26]. Given that NK cells also play a key role in priming the adaptive immune system to respond to invading pathogens or to tailor their immunogenicity towards specific vaccines[27], it is important to investigate the maturation trajectory of NK cells in tandem with T cell immune ontogeny. Whether early life exposure to HIV/ARV shapes the maturation and/or diversification of NK cells remains unknown.

Here, we comprehensively investigated the relationship between adaptive and innate immunophenotypes along with TCR diversity and the ability to mount an antibody response to pertussis and rotavirus vaccination. Given the heightened infectious morbidity risk in iHEU, we hypothesised that immune ontogeny is associated with a narrowing TCR repertoire over time in parallel with expanded NK cell subsets in iHEU relative to iHUU. We mapped the immune trajectory of NK and T cell phenotypic clusters in the first 9 months of life and associated this with TCR diversity and antibody titres. Using this approach, we show divergent adaptive immune maturation in iHEU relative to iHUU occurring from 3 months of life with earlier phenotypic differences evident in NK cells. Our data show a narrowing and persistent skewing of the TCR repertoire from birth in iHEU that precedes altered CD4+ and CD8+ T cell memory development. We also show that specific NK cell clusters at birth can predict the magnitude of pertussis and rotavirus vaccine-induced antibody response and highlights the importance of innate immunity in early life.

## Results
### Immune cell transition from birth to 9 months of age
Our infant cohort consisted of 36 infants (iHEU=40 and iHUU=16) and having similar population characteristics including birth weight and gestational age, with the exception that the mothers of iHEU were noted to be significantly older (Table S1). To characterise the immune ontogeny in our infant cohort (Table S1 and Fig. S1), we first analysed the overall immunological trajectories of innate and adaptive immune cells subsets in the first 9 months life. Matched PBMC samples collected at birth and at week 4, 15 and 36 as well as some matching cord blood mononuclear cells were analysed for immunophenotypic changes using a mass cytometry antibody panel (Table S2).

Unsupervised cell clustering using the live singlet cell population (Fig. S2A) from all infant samples was used to describe the trajectories of myeloid and lymphoid lineages in infants. Using this approach, we could identify 11 cell clusters (Fig. S2B and S2C), consisting of B cells (cluster 1), T cells (CD4+ T cluster 2 and CD8+ T clusters 3, 4, and 9), monocytes (cluster 8), NK cells (cluster 10), and NKT-like cell (cluster 11). The three CD8+ T cell clusters (3, 4 and 9) were delineated as naïve (13.9%, CD45RA+CD27+CCR7+), a small effector memory population (0.07%, EM: CD45RA+CD27+CCR7-) and effector cells (4.4%, Eff: CD45RA+CD27-CCR7-). We also identified a lineage negative cluster (3.2%) and one monocyte-like cluster expressing HLA-DR only (1.2%). Although we employed computational doublet removal from our clustering analysis, a minor miscellaneous population (0.65%) co-expressing typically mutually exclusive lineage markers was identified (Fig. S1A and S1B). This small population is shown as a central cluster 7 in the dimensional reduction of the immune cell clusters (Fig. S2C) and was excluded from downstream analysis.

When we examined the remaining clusters over time, we observed an age dependent segregation of cells. The proportions of cell clusters changed over 36 weeks of life, where monocytes ($\rho = -0.4$, $p < 0.001$) and CD4+ T cells ($\rho = -0.36$, $p < 0.001$) diminished with age, B cells ($\rho = 0.66$) and effector memory CD8 + T cells ($\rho = 0.54$) were positively correlated to infant age (Fig. S2D and S2E).

### Distinct temporal maturation trajectories of NK cells stabilising at 4 months coinciding with rapid CD8 + T cell divergence
Since our mass cytometry antibody panel (Table S2) was mainly focused on assessing ontological changes of NK cells and T cells, we then manually gated on CD4+ and CD8+ T cells from the CD3+ T cell population and targeted the NK cell population using lineage-exclusion manual gating (Fig. S2A). The median marker expression for each sample was used to compute multidimensional scaling (MDS) coordinates, which revealed that intra-individual variability amongst the infants was associated with age (Fig. 1A–F). The grouping of the samples by marker expression displayed a converging trajectory along the MDS2-axis for NK cells although stabilizing from week 15 (Fig. 1D). The CD4+ displayed a gradual maturation transition from birth to week 36, consistent with previous findings showing neonatal to adult immunological transition[28]. CD8+ T cells, however, revealed a divergent trajectory along the MDS1-axis (Fig. 1D), with a significant expansion by week 15 (Fig. 1E).

We then proceeded to identify the maturation trajectories of the different subsets within each cell population by performing unsupervised cell clustering based on marker expressions. Across all time points, we could identify 10 NK cell clusters, 13 CD4+ T cells and 7 CD8+ T cells. (Fig. 1G–L, Tables 1–3 and Fig. S3). Principal Component Analysis (PCA) using centred log-ratios of the relative cluster abundances for each sample also demonstrated grouping of the infants by their respective age (Fig. 1H, J, L). NK cell cluster 8 (CD56+CD16-NKG2A+CD45RA+Perorin+) was more abundant at earlier time points (birth and week 4), while cluster 5 (CD56loCD16+CD57+CD45RA+Perforin+) was significantly abundant at later time points (week 15 and 36) demonstrating a typical NK cell maturation trajectory (Figs. 1H and S3A). Similarly for T cell populations, CD4+ and CD8+ clusters reflected the immunological trajectory shown in Fig. 1B, C, with the terminally differentiated clusters: CD4+ cluster 3 (CD45RA+CD57+Perforin+PD-1+) and CD8+ clusters 1

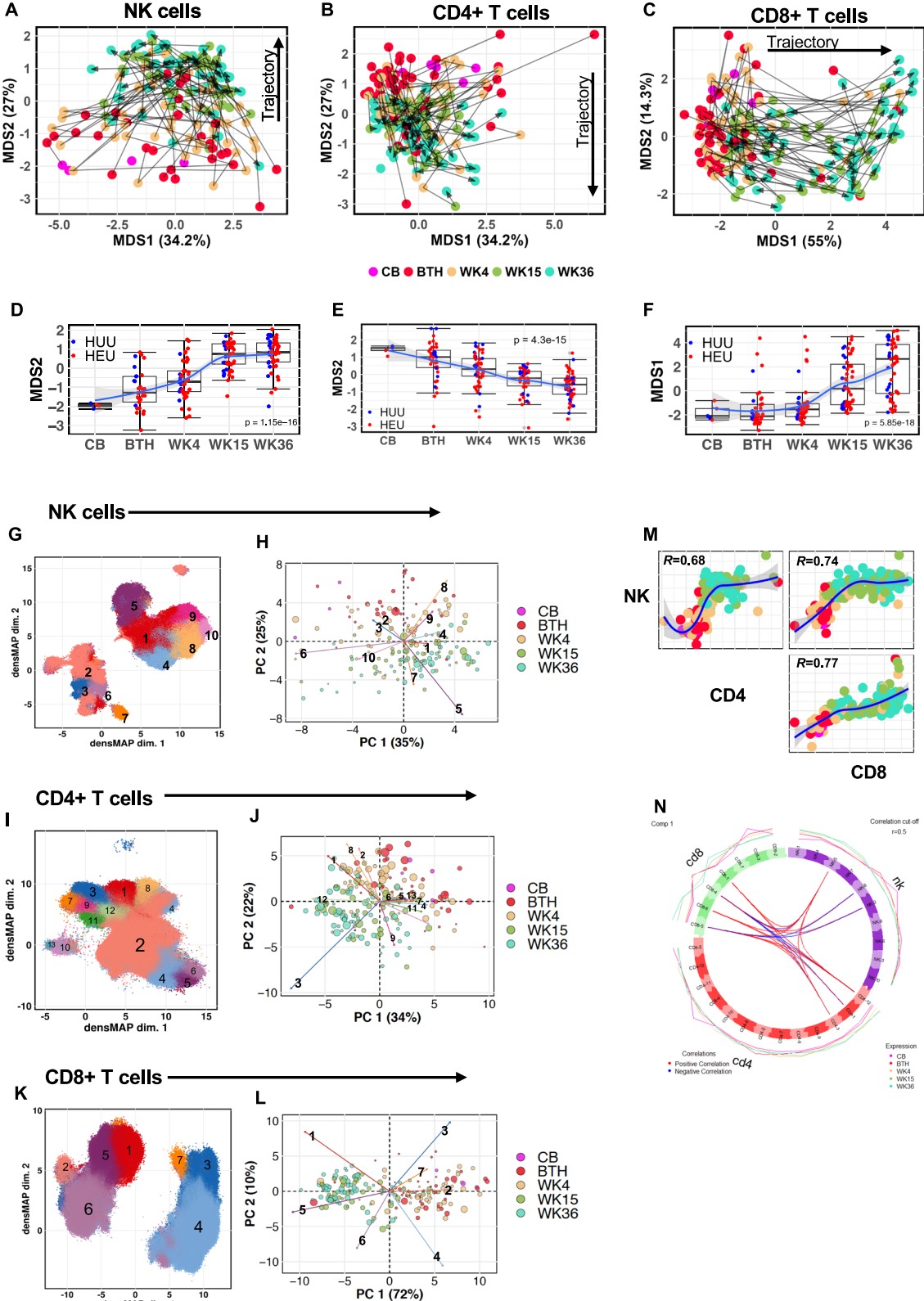

(CD45RA⁺CD57⁺Perforin⁺CCR4⁺) and 5 (CD45RA⁺CD57⁺Perforin⁺) increasing with infant age (Fig. 1I–L, S3B and S3C).

We further interrogated the compositional relationship amongst the different cell clusters by integrating the 3 data sets derived from NK cells and CD4⁺ and CD8⁺ T cells and performed a block partial least square regression with discriminant analysis using DIABLO (Data Integration Analysis for Biomarker discovery using Latent

cOmponents)[29]. Using this analytical approach, we could observe the strong correlation between the maturation of CD4⁺ and CD8⁺ T cells with advancing infant age (Spearman's rank correlation, Fig. 1M). Moreover, the converging maturation trajectory of NK cells, which stabilised at week 15, coincided with the rapid expansion of CD8⁺ T cells (Fig. 1M). This suggests maturational regulation between the cytolytic cells during the neonatal - infant transition as demonstrated

**Fig. 1 | Immunophenotypic trajectory of infant NK cells and T cells in the first 9 months of life. A–C** Age related maturation trajectory of NK cells and CD4[+] and CD8[+] T cells depicted using multidimensional scaling (MDS) coordinates derived from median marker expressions for each infant samples collected in cord blood (CB, $n = 5$) and at birth (BTH, $n = 44$) and weeks (WK) 4 ($n = 53$), 15 ($n = 52$) and 36 ($n = 53$). **D–F** Boxplots summarising MDS coordinate associated with infant age for NK cells and CD4[+] and CD8[+] T cells. Boxplot central line depicts the median, the upper and lower lines represents the 75th and 25th percentile respectively, and the whiskers mark the 1.5 times the 75th and 25th percentile boundary, ($p$ = Kruskal Wallis). Summary trendline shows local regression as determined by local estimated scatterplot smoothing (LOESS) with 95%CI error bands. **G, I, K** Uniform manifold approximation and proximation (UMAP) with density preservation showing dimensional reduction of FlowSOM clusters of NK cells and CD4[+] and CD8[+] T cells, respectively. **H, J, L** Age related phenotypic composition of NK cells and CD4[+] and CD8[+] T cells depicted as principal component (PC) coordinates of centred log-odd ratios of the FlowSOM clusters for each infant samples and arrows indicating contribution of each cell cluster in the scatter of PC components. **M** Spearman's correlation of PC coordinates associated with immune trajectory for NK cells, and CD4[+] and CD8[+] T cells. Summary trendline shows local regression as determined by LOESS with 95%CI error bands. **N** Circoplot showing correlations among immune cell clusters derived from NK cells, and CD4[+] and CD8[+] T cells.

by the negative correlation between NK cell cluster 8 (Perforin[+]CD38[+]NKG2A[+]Nkp30[+]NKp46[+]Siglec-7[+]FcεRIγ[+]) and CD8[+] T cell cluster 5 (Perforin[+] CD57[+]CD45RA[+]DNAM1[+]LILRB1[+]2B4[+]), while CD8 + T cell cluster 5 was positively correlated to NK cell cluster 5 (Perforin[+]CD45RA[+]CD38[+]FcεRIγ[+]LILRB + CD57[+]).

Overall, these temporal phenotypic changes in cell clustering illustrate the archetypal progression and transition of infant immunity from neonatal to an adult-like immune phenotype by 9 months of age, being characterised by a shift from innate myeloid cells to an increase in lymphoid adaptive immune cells[30].

## Memory maturation of CD4 and CD8 T cells is disrupted by HIV exposure after three months of life

In our previous analysis we observed altered expansion of T regulatory cells among iHEU relative to iHUU counterparts[17]. We were thus interested in determining the extent of maternal HIV infection in modulating the trajectory of infant immune maturation. PCA components derived from the centred log-ratios of the relative cluster abundances for each cell subsets were compared between iHEU and iHUU to determine compositional differences. There were statistically significant differences at weeks 15 and 36 for CD4[+] T cells and at week 36 for CD8[+] T cells (Fig. 2B, C). Differential abundance testing using generalised linear mixed model revealed that the divergent CD4[+] T cell phenotypes in iHEU were due to lower frequencies of closely related activated and proliferating (HLA-DR[+]Ki67[+]) clusters 4 & 11, and memory differentiated cytolytic (PD-1[+]CD57[+]Perforin[+])[31,32] clusters 5-9 compared to iHUU at week 15, with the differentiated cytolytic clusters 5, 7 and 9 remaining significantly lower in iHEU at week 36 (Fig. 2D, E). The CD4[+] Th2-like cluster 1 (CCR4[+]CD27[+]CD127[+]) was, however, significantly higher in iHEU compared to iHUU at week 15 although by week 36 was no longer significantly elevated (Fig. 2D, E). We further noted that the proportion of Th2-like cluster was also higher in iHEU compared to iHUU at week 4, albeit not significant following correction for multiple comparisons (Fig. S4A).

For CD8[+] T cells, the compositional differences were partly driven by lower mean frequencies for clusters 2 (NKT-like cells: CD56[+]CD16[+]Nkp30[+]NKp46[+]NKG2A[+]Perforin[+])[33], 3 (Naïve-like activated cell: CD45RA[+]CD27[+]CCR7[+]CXCR3[+]CCR4[+]HLA-DR[+]) and 7 (proliferating EM: Ki67[+]CD45RA[+]CD27[+]CCR7[-]Perforin[+]) in iHEU compared to iHUU at week 15 (Fig. 2I). By week 36, only the mean frequencies of clusters 2 (NKT-like cells) and 3 (Naïve-like cells) remained significantly lower in iHEU compared to iHUU (Fig. 2F, G).

The composition of NK cells differed by HIV-exposure only at week 4 (Fig. 2A), partly driven by higher frequencies of the differentiated (CD56[lo]NKG2A[-]CD57[lo/+])[34] NK cell cluster 1 (CD56[lo]CD16[lo]Perforin[+]CD38[+]CD45RA[+]FcεRIγ[+], $p = 0.04$, $p$.adj = 0.2) and cluster 5 (CD56[lo]CD16[+]Perforin[+]CD57[+]CD45RA[+]CD38[+)]) ($p = 0.03$, $p$.adj = 0.2) in iHEU compared to iHUU (Fig. S4B), although these difference were not significant after correcting for multiple testing.

These findings show that HIV/ARV exposure sequentially disrupts immune trajectory over time after birth beginning with subtle changes in innate NK cells and delaying memory T cell maturation.

## Early T Cell Receptor repertoire skewing in iHEU

To relate altered T cell memory maturation trajectories to T cell repertoire changes, we characterised TCRβ clonotypes in sorted naïve and memory T cells from matching samples used to define memory lineage in our CyTOF panel (Fig. S1). Each PBMC sample was sorted into four T cell subsets: naïve (CD45RA[+]CD27[+]CCR7[+]) CD4[+] and CD8[+] cells and total memory (CD45RA[-/+] CD27[-/+]CCR7[-/+]) T cells (Fig. S5A and Data S1). The sorted cell fractions were subjected to bulk sequencing of the TCR Vβ locus which enabled identification of TCR clones in 885 samples. There were 24 samples that had either fewer than 10 clones or the number of unique clones were greater than the number of cells sequenced, and these were removed from downstream analysis (Fig. S1). Of the remaining 861 samples included in the study analysis, there was a total of 238,092 reads (range: 11-5451), which differed between iHEU and iHUU for naïve CD4[+] T cells at weeks 36 (Data S1). The number of unique clones identified per T cell subset positively correlated with the total number of reads (Fig. S5B). The median number of unique clones per T cell subset, however, was higher in memory T cells for iHUU compared to those of iHEU at birth, and for naïve CD4[+] T cell a similar difference was observed at week 36 (Data S1). CDR3 lengths of the clonotypes were evenly distributed across infant age and between iHEU and iHUU (Fig. S5C).

We assessed the TCRβ repertoire during infancy by calculating the inverse Simpson diversity index and Chao1 richness scores and noted that the overall TCRβ repertoire remained relatively stable over the first 9 months (Fig. S5D and S5E). However, when the TCRβ diversity scores were stratified by infant HIV exposure status, the TCRβ diversity for iHUU memory CD4[+] T cells significantly decreased over time ($p = 0.032$, Fig. S6A). Moreover, iHEU memory TCRβ clonotypes had relatively lower diversity compared to iHUU at birth (Fig. 3A). Similarly, richness (the number of unique TCRβ clones) was significantly lower in

## Table 1 | Phenotypic description of NK cell clusters identified by unsupervised cell clustering

| Cluster | Phenotype | (%) |
|---|---|---|
| (1) | (CD56[lo]CD16[lo]) Perforin+ CD38+ CD45RA+ FcεRIγ + | 30.5 |
| (2) | (CD56[-]CD16[-]) DNAM1+ | 20.7 |
| (3) | (CD56[-]CD16[+]) FcεRIγ + | 1.8 |
| (4) | (CD56[lo]CD16[+]) Perforin+ CD45RA+ CD38+ 2B4+ FcεRIγ+ | 11.7 |
| (5) | (CD56[lo]CD16[+]) Perforin+ CD45RA+ CD38+ FcεRIγ + LILRB+ CD57+ | 12.4 |
| (6) | (CD56[lo]CD16[lo]) FcεRIγ+ DNAM1+ HLA-DR+ CXCR3+ CCR4+ | 1.4 |
| (7) | (CD56[-]CD16[+]) CD38+ CD39+ CD27+ Ki67+ | 0.9 |
| (8) | (CD56[+]CD16[+]) Perforin+ CD38+ CD45RA+ DNAM1+ 2B4+Nkp30+ NKp46 + NKG2A+ Siglec-7+ FcεRIγ+ | 15.0 |
| (9) | (CD56[bri]CD16[-]) DNAM1+ NKG2D+Nkp46 + NKG2A+ | 4.9 |
| (10) | (CD56[+]CD16[+]) Perforin+ CD38+ CD45RA+ LILRB1+ NTBA+ 2B4+ Nk30+ NKG2D+ HLA-DR+ CCR4+ CCR7+ PD-1+ TIGIT+ NKg2C+ | 0.7 |

**Table 2 | Phenotypic description of CD4⁺ T cell clusters identified by unsupervised cell clustering**

| Cluster | Phenotype | (%) |
|---|---|---|
| (1) | CCR4+ CD27+ CD127+ (Th2) | 5.07 |
| (2) | CD38+ CD45RA+ CD27+ CCR7+ CD127+ (Naïve) | 84.1 |
| (3) | CD45RA+ CD57+ Perforin+ PD-1+ (Cytotoxic TD) | 1.3 |
| (4) | CD38+ HLA-DR+ CD45RA+ CD27+ CCR7+ (Activated cells) | 4.7 |
| (5) | CD27+ CD38+ CD45RA+ CCR7+ CCR4+ CCR6+ CXCR3+ TIGIT+ PD-1+ HLA-DR+ CD69+ (Th1/17 Activated) | 0.7 |
| (6) | CD27+ CD38+ CD45RA+ CCR7+ CCR4+ CCR6+ CXCR3+ TIGIT+ PD-1+ CD57+ CD39+ (Th1/Th17) | 0.2 |
| (7) | CD27+ Perforin+ CD45RA+ CCR7+ CD38+ TIGIT+ Ki67+ | 0.3 |
| (8) | CD25+ CD39+ TIGIT+ CCR4+ PD-1+ (Treg cells) | 0.9 |
| (9) | CD27+ CD38+ CD45RA+ CCR7+ Perforin+ Ki67+ CD25+ | 0.3 |
| (10) | CD27+ CD38+ CD45RA+ CCR7+ Perforin+ Ki67-HLA-DR+ CD69+ CD39+ | 0.1 |
| (11) | Perforin+ Ki67+ CD27+ CD38+ CD45RA+ CCR7+ CD127+ | 0.2 |
| (12) | CD27+ PD-1+ CD38+ | 2.07 |
| (13) | CD39+ CD45RA+ | 0.1 |

**Table 3 | Phenotypic description of CD8⁺ T cell clusters identified by unsupervised cell clustering**

| Cluster | Phenotype | (%) |
|---|---|---|
| (1) | Perforin+ CD45RA+ CD38+ HLA-DR+ CCR4+ Siglec-7+ CD57+ (TD) | 7.5 |
| (2) | CD56+ CD16+ Perforin+ FcεRIγ+ NkpK6+ NKG2A+ (NKT-like) | 0.4 |
| (3) | CD45RA+ CD27+ CCR7+ CD127+ HLA-DR+ CXCR3+ CD38+ | 2.7 |
| (4) | CD45RA+ CD27+ CCR7+ CD127+ CD38+ (Naïve) | 55.3 |
| (5) | Perforin+ CD57+ CD45RA+ DNAM1+ LILRB1+ 2B4+ | 15.6 |
| (6) | Perforin+ CD38+ 2B4+ NTBA+ | 18.4 |
| (7) | Ki67+ CD45RA+ CD27+ NKp30+ KIR2DL1+ | 0.1 |

iHEU relative to iHUU for both memory CD4⁺ and CD8⁺ T cells at birth (Fig. 3B). This remained statistically significant for memory CD4⁺ T cells at week 4 (Fig. 3B).

In contrast to TCRβ of memory T cells, in which significant differences were found early life, within the naïve T cell compartment; TCRβ diversity was significantly lower in iHEU compared to iHUU at week 15 and 36 for CD4⁺ T cells, and at birth and week 36 for CD8⁺ T cells (Fig. 3A). Significant differences in TCRβ richness for naïve T cells were partially complementary to the differences observed for TCRβ diversity, being lower in iHEU compared to iHUU at birth and week 36 for CD4⁺ T cells and only at birth for CD8⁺ T cells (Fig. 3B).

We also observed differences in the overall structural overlap, as measured by Jaccard indices, of the memory CD4⁺ TCR repertoire between iHEU and iHUU at birth, although this difference was less prominent at later time points (Fig. S6B). In contrast, the TCRβ structural overlap for naïve CD8⁺ T cells was lower in iHEU compared to iHUU only at week 36 (Fig. S6B). Differences in Vβ gene usage were observed between iHEU and iHUU with most genes being used more frequently in iHEU (Fig. S6C). Together, these results suggest expansion of specific TCR clones among iHEU, resulting in skewing of the TCRβ repertoire relative to iHUU and which likely occurred *in utero*.

Since the naïve TCRβ clonotypic differences coincided with the lower memory differentiated T cell clusters (Fig. 2), we next wished to determine if there was a relationship between TCRβ diversity and the clusters of CD4⁺ and CD8⁺ T cells identified. Spearman's rank correlation between T cell clusters and the inverse Simpson TCRβ diversity scores for naïve and memory CD4⁺ and CD8⁺ T cells were measured for infants with paired TCRβ and mass cytometry data at all time points.

Here, the TCRβ diversity scores derived from sorted naïve T cells in iHEU were positively correlated to the frequencies of CD4⁺ T cell clusters (4–6, 9–11) and CD8⁺ T cell clusters (2,3,7) (Fig. 3C).

It is noteworthy that the observed T cell clusters that were correlated to the naïve TCRβ diversity included the HLA-DR⁺Ki67⁺CD4⁺ T cell clusters (4 & 11), memory differentiated CD4⁺ T cell clusters (5,6, 9 & 11) and the CD8⁺ T cell clusters (2,3 & 7) that were significantly lower amongst iHEU at week 15 and 36 (Fig. 2D–G). Relative to the naïve CD4⁺ and CD8⁺ T cell clusters, the T cell clusters that were correlated with TCRβ displayed varying expression levels of PD-1 suggesting TCR activation (Fig. S3B and S3C)[35,36]. For the proliferating activated CD4⁺ T cells (cluster 4 & 11) and the terminally differentiated cytolytic CD4⁺ T cells (cluster 5 & 6), these correlations were evident from birth up until week 15. Although a similar trend was observed for the T cells clusters and TCR diversity derived from sorted memory T cells, the observed correlations were not statistically significant after correcting for multiple comparisons (Fig. S6D).

Collectively, our findings show skewing of the TCR repertoire in iHEU relative to iHUU from birth and the early increased T cell clonality preceding T cell memory differentiation was a hallmark finding of HIV/ARV exposure.

## Predicted TCR recognition of epitopes found only in iHUU

To understand the predicted antigen specificities of the TCR clonotype differences between iHEU and iHUU, we used GLIPH2 (grouping of lymphocyte interactions by paratope hotspots)[37,38]. TCR specificities from sorted naïve CD4⁺ T cells in iHUU appeared enriched with specificities for cytomegalovirus (CMV), Hepatitis C virus (HCV), SARS-Cov2 and HIV-1 (Fig. 3D). Enrichment of Influenzae A specific clones was more apparent in memory CD4⁺ T cells at the week 36. For memory CD4⁺ T cells at week 4, the shared specificity groups in iHUU were enriched for CMV clones. Similarly, for CD8⁺ T cells, there were more specificities in the naïve compartment at both earlier and later time points with less predictions in memory CD8⁺ T cells. TCR specificities in sorted naïve CD4⁺ and CD8⁺ cells from iHUU appeared consistent throughout 36 weeks, whereas specificities in the CD4⁺ T cell memory compartment fluctuated and appeared predominantly at 4 and 36 weeks (Fig. 3D). Conversely for iHEU sorted T cell populations, no enriched TCR specificities in either naïve or memory CD4⁺ or CD8⁺ cells at any of the time points were predicted.

## NK cell clusters are the strongest predictors of antibody responses

To relate how early life NK cell and T cell clusters associate with functional immune outcomes in infants included in this study, antibody responses were measured against acellular pertussis vaccination from birth to 36 weeks and to rotavirus at 36 weeks in matching infants used for immunophenotyping by mass cytometry (Fig. S1). We initially hypothesised that iHEU would have lower IgG and IgA responses compared with iHUU after vaccination with routine administration of these vaccines, which may explain a mechanism for higher infection rates among iHEU. The median anti-pertussis IgG titres were lower in iHEU compared to iHUU at birth and at week 4 (pre-vaccination; Fig. 4A). However, at week 15 (1 week post 3rd vaccine dose), iHEU showed a significantly higher anti-pertussis IgG response relative to iHUU, persisting until week 36 (Fig. 4A). No significant differences were observed in anti-rotaviral IgA levels or ability to neutralise rotavirus between iHEU and iHUU (Figs. 4B and S7A).

To determine contemporaneous cell clusters that were related to these antibody responses, we correlated the frequencies of cell clusters at weeks 15 and 36 with anti-pertussis IgG and anti-rotavirus IgA titres. CD4⁺ T cell cluster 1 (Th2-like cells) associated with antibody production[39], was significantly correlated to pertussis antibody titres at week 15 ($\rho = 0.44$, $p$.adj = 0.048; Fig. 4C). Although a similar trend was observed for both iHEU and iHUU, these correlations were not

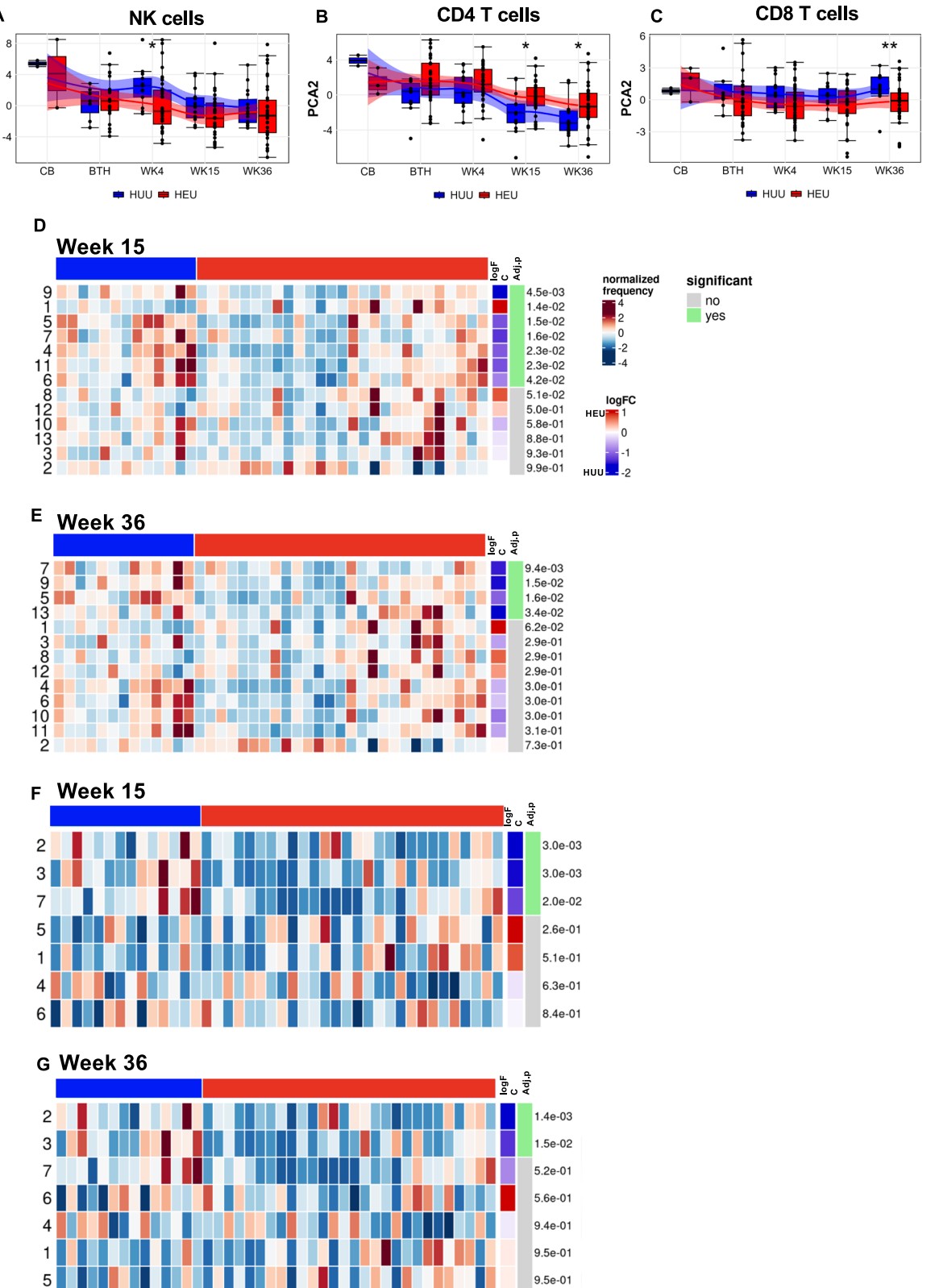

**Fig. 2 | Divergent T cell memory differentiation in HIV-exposed uninfected infants (iHEU) compared to HIV-unexposed uninfected infants (iHUU).**
**A–C** Boxplots comparing PC coordinates derived from centred log-odd ratios of FlowSOM NK cell, and CD4⁺ and CD8⁺ T clusters between iHEU and iHUU at birth (BTH), weeks (WK) 4, 15 and 36. Boxplot central line depicts the median, the upper and lower lines represents the 75th and 25th percentile respectively, and the whiskers mark the 1.5 times the 75th and 25th percentile boundary (*$p < 0.05$, **$p < 0.01$; Two-sided Wilcoxon). Summary trendline shows local regression as

determined by local estimated scatterplot smoothing (LOESS) with 95%CI error bands. **D, E** Generalised linear mixed model (GLMM) comparing the abundances of CD4⁺ T cell clusters between iHEU and iHUU at weeks 15 and 36. **F, G** GLMM comparing the abundances of CD8⁺ T cell clusters between iHEU and iHUU at weeks 15 and 36. Positive log Fold change (FC) signified cell cluster frequencies were higher in iHEU compared to iHUU. False discovery rate (FDR) was used to correct for multiple comparison and adjusted $p$ values are presented.

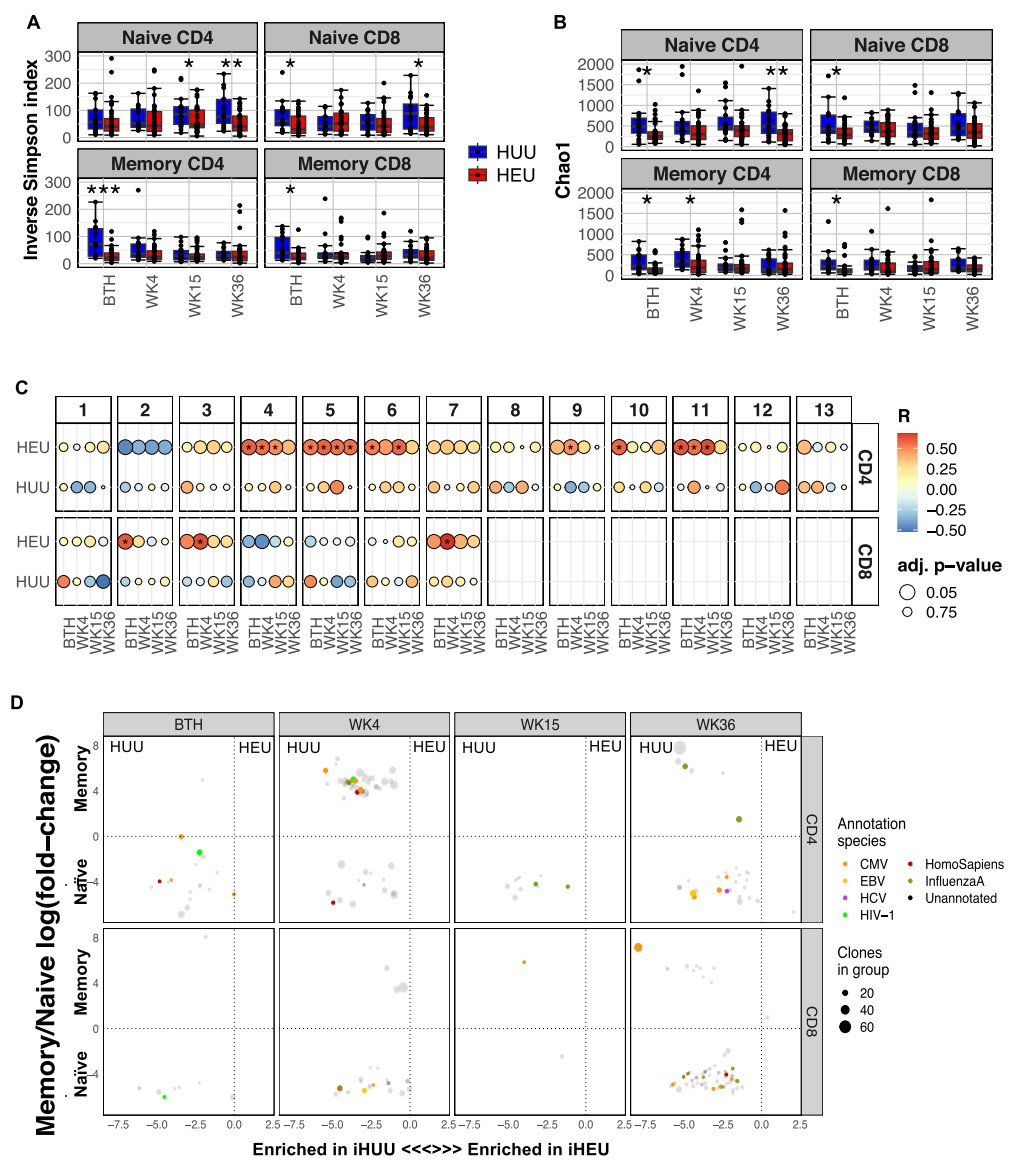

**Fig. 3 | Premature CD4+ and CD8 + T cell receptor (TCR) repertoire skewing in HIV-exposed uninfected infants (iHEU) relative to HIV-unexposed uninfected infants (iHUU).** **A** Boxplots comparing Inverse Simpson TCR diversity scores between iHUU and iHEU at birth (BTH) and weeks (WK) 4, 15 and 36. **B** Boxplots comparing Chao1 TCR clonotype richness between iHUU and iHEU at BTH and WK4, 15 and 36. All boxplots used the standard Tukey's representation with the central line depicting the median, the upper and lower lines represent the 75th and 25th percentile respectively, and the whiskers mark the boundary at 1.5 times of the 75th and 25th percentile ($*p < 0.05$, $**p < 0.01$ and $***p < 0.001$; Two-sided Wilcoxon). **C** Spearman's rank correlation between naïve CD4$^+$ and CD8$^+$ T cell Inverse Simpson scores and frequencies of CD4$^+$ and CD8$^+$ T cell clusters respectively. *adjusted $p < 0.05$, FDR used to correct for multiple comparisons. **D** GLIPH analysis showing antigen specificity that were significantly enriched in iHUU relative to iHEU and in naïve relative to memory CD4$^+$ and CD8$^+$ T cell subsets. Sample size for each group is shown in Fig. S1.

statistically significant after adjusting for multiple comparisons (Fig. S7B). At week 36, there was an inverse correlation between pertussis antibody levels and CD4$^+$ T cell cluster 11 (HLA-DR$^+$Ki67$^+$), terminally differentiated clusters 6,7, and 9 (PD-1 + CD57+Perforin + ), and clusters 10 and 13 (CD38$^+$CD39$^+$), and also the CD8$^+$ T cell cluster 7 (Proliferating EM: Ki67$^+$CD45RA$^+$CD27$^+$CCR7$^-$). Of note, there was no significant contemporaneous correlation between the NK cell clusters and pertussis antibody titres (Figs. 4C and S7B). In addition, the levels of anti-rotavirus IgA at week 36 were not correlated to any of the identified NK cell and T cell clusters (Fig. 4D).

We next assessed whether immune cell compositions at birth could predict vaccine outcomes during infancy. We built a prediction model by grouping all infants (iHEU and iHUU) into those that had a positive pertussis-specific IgG titre (≥0.469 OD$_{Abs}$) denoted as responders and those that were non-responders (<0.469 OD$_{Abs}$). Most of the infants at birth and week 4 (89.7% and 94.1%, respectively) were determined to be non-responders as expected, since this was prior to vaccination. At week 15 and 36 of age, 46.4% and 34.0% infants respectively were determined to be responders (Fig. 4A).

Multivariate unbiased variable selection using MUVR[40] was performed to determine early life immune composition that could distinguish later pertussis vaccine responders from non-responders. The abundances of cell clusters with low misclassification MUVR scores of pertussis vaccine response at weeks 15 and 36 (Fig. S8), were then used to perform a partial least squares discriminant analysis (PLS-DA) with the pertussis IgG titres at week 15 and 36 as response variables. The

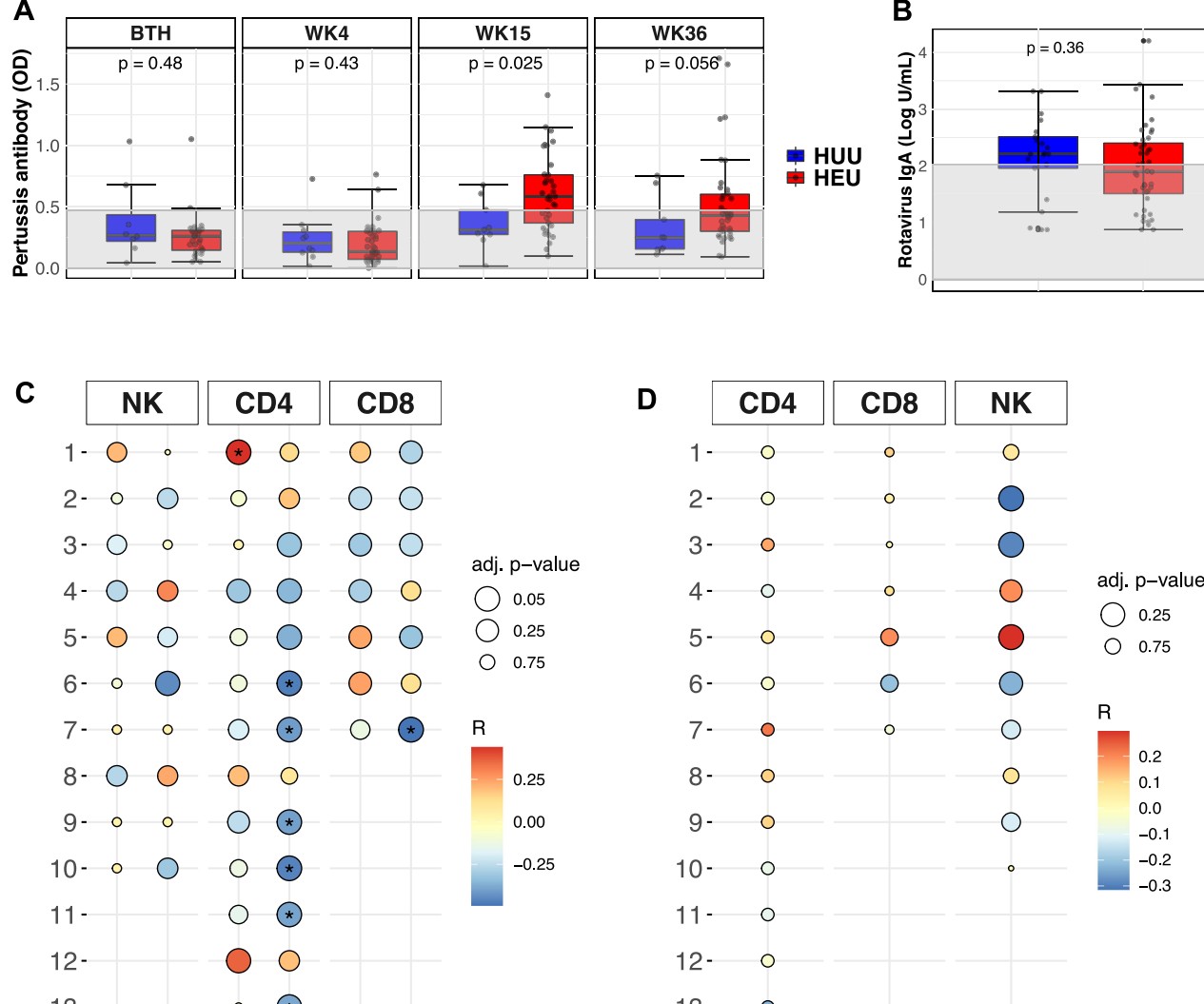

**Fig. 4 | Association of vaccine antibody responses to immune cell phenotypes.**
**A** Comparing IgG levels against pertussis between HIV-exposed uninfected infants (iHEU) and HIV-unexposed uninfected infants (iHUU), grey shaded area indicate threshold IgG levels for protective pertussis vaccine response. **B** Boxplots comparing rotavirus specific IgA titres between iHEU and iHUU at week 36. All boxplots used the standard Tukey's representation with the central line depicting the median, the upper and lower lines represent the 75th and 25th percentile respectively, and the whiskers mark the boundary 1.5 times of the 75th and 25th percentile (*$p < 0.05$, **$p < 0.01$ and ***$p < 0.001$; Two-sided Wilcoxon). **C**, **D** Spearman's correlation of FlowSOM clusters abundances for NK cells, CD4+ and CD8+ T cells to anti-pertussis IgG and anti-rotavirus IgA respectively measured at week 15 and 36. *adjusted $p < 0.05$, FDR used to correct for multiple comparisons.

latent variables (LV) of the cell clusters as determined by PLS-DA were then used to compute area under the curve (AUC) of the receiver operating curve (ROC) distinguishing pertussis responders from non-responders (Figs. 5A and S7C). This approach revealed that compared to T cells, the NK cell cluster compositions at birth and at week 4 were superior predictors of the pertussis IgG vaccine response at week 15 (Fig. 5A, at birth: AUC = 0.93 and $p = 0.005$; at week 4: AUC = 0.89 and $p = 0.0003$). NK cell cluster 1 (CD56loCD16lo Perforin+CD38+CD45RA+FcεRIγ+) and cluster 3 (CD56−CD16+ FcεRIγ+) measured at birth and week 4 were consistently associated with responders to pertussis vaccine at week 15, while cluster 8 (CD56+CD16+Perforin+CD38+CD45RA−DNAM1+2B4+Nkp30+NKp46+NK-NKG2A+Siglec-7+FcεRIγ+) was associated with non-responders to pertussis vaccine (Fig. 5B).

The cell clusters derived from CD4+ or CD8+ T cells were not predictive of IgG pertussis vaccine response at weeks 15 and 36

(Fig. 4D and S7C), for CD4 clusters at birth: AUC = 0.73, $p = 0.064$; for CD8 clusters: AUC = 0.69, $p = 0.12$; at week 4: CD4 clusters AUC = 0.70, $p = 0.055$; CD8 clusters AUC = 0.70, $p = 0.053$.

Using the same approach for rotavirus responses, we used an arbitrary threshold of overall median concentrations for IgA responses to dichotomize infants into those that had concentrations below the median (low-responders) and those above the median (high-responders; Fig. 4B). In iHUU, 66.7% were high rotavirus vaccine responders compared to 45.5% iHEU ($p = 0.11$). We subsequently determined which immune cell clusters at birth and week 4 were predictive of IgA responses measured at week 36 using MUVR (Fig. S9). PLS-DA analysis using the MUVR selected cell clusters revealed that NK cells at birth were highly predictive of an IgA rotavirus response at week 36 (Fig. 5C, AUC = 0.92, $p = 0.0007$). Similar to the anti-pertussis response, NK cell cluster 1 was also associated with high anti-rotavirus IgA responses, while NK cell cluster 8 was associated with low anti-rotavirus IgA

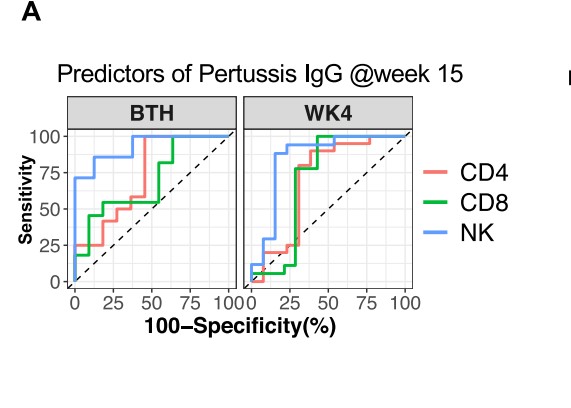

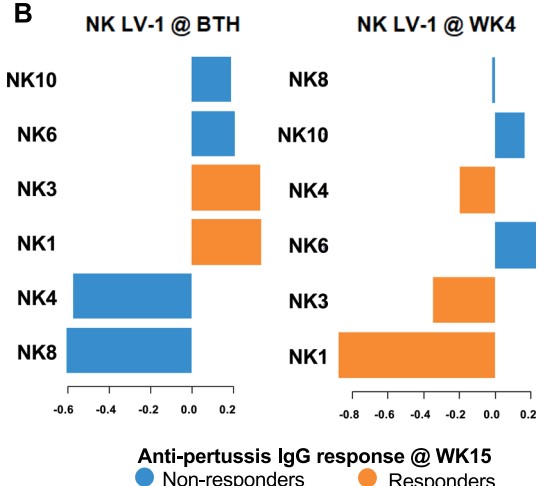

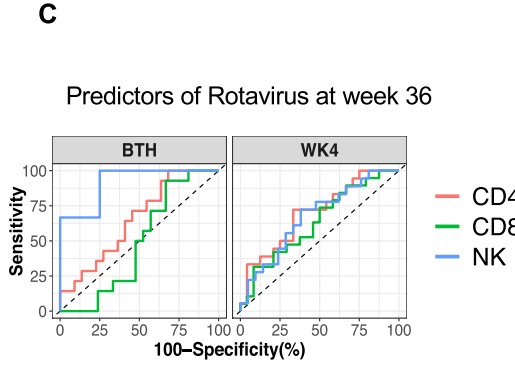

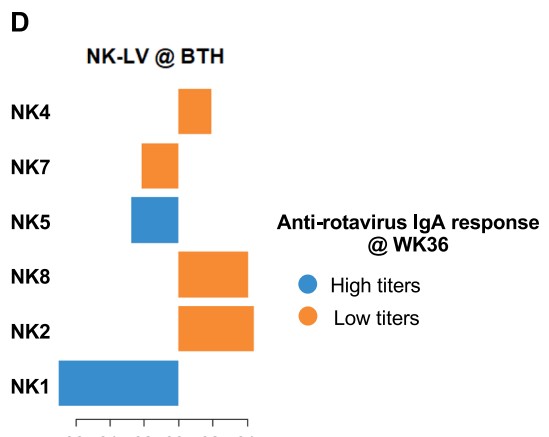

**Fig. 5 | Early-life immune cell compositions predictive of antibody responses post-vaccination. A**, **C** Summary of ROC analysis using the latent variable axis-1 derived from partial least square discriminate analysis (PLS-DA) of NK, CD4 and CD8 T cell clusters determined to be best predictors at birth and week 4 of pertussis antibody responses at weeks 15 and rotavirus antibody response at week 36. **B**, **D** Loading variables from PLS-DA showing pre-vaccine cell clusters associated with good or poor antibody responses against pertussis and rotavirus vaccine at week 15 and 36 respectively. Sample size for each group is shown in Fig. S1.

(Fig. 5D). CD4+ or CD8+ T cell clusters were less accurate in predicting anti-rotavirus IgA response at week 36 (Fig. 5C, for CD4 clusters at birth: AUC = 0.65, $p = 0.13$; for CD8 clusters: AUC = 0.49, $p = 0.89$; at week 4: AUC = 0.7, $p = 0.027$; for CD8 clusters: AUC = 0.62, $p = 0.17$).

Collectively, these data show that specific NK cell phenotypes at birth, and in the first month of life, can predict vaccine-induced antibody responses to acellular pertussis and rotavirus at post-vaccination, regardless of infants being exposed to HIV. These data highlight the potential association between an early innate immune response with later T-cell dependent vaccine-induced antibody responses.

## Discussion

Herein, we describe immune maturation in the first 9 months of life and how HIV/ARV exposure alters this ontological trajectory. Our findings show ordered immune changes during infancy, typical of the transition from innate-like and naïve neonatal immunity towards memory differentiated adaptive infant immunity[30,41]. HIV exposure altered the immunological compositions of NK cells and T cells at different stages during infancy, with the T cell memory maturation being significantly impacted later in life. The impaired T cell memory maturation among iHEU could, in part, be due to diminished naïve TCR clonal pools observed in iHEU relative to iHUU. The skewing of TCR

clonality most likely initiated *in utero*, as differences between infant groups were observed as early as birth, as previously shown[42,43]. We further showed that pre-vaccine NK cell composition could predict vaccine-induced antibody levels in infants better than T cells and is consistent with studies reporting baseline immunological signatures that are predictive of vaccine responses[44].

Although differentiated T cells are detected as early as 12−14 weeks in gestation[45,46], neonatal immunity is predominantly naïve and characterised by the abundance of CD45RA+ T cells[7]. T cell memory expansion during infancy is therefore essential in protecting against pathogens[47]. Here we observed that the overall trajectory of T cell phenotypic composition in iHEU diverged from that of iHUU, particularly after week 15 of life. Both CD4+ and CD8+ T cell compartments in iHEU had lower frequencies of activated cells and memory differentiated cells expressing either PD-1 and/or perforin compared to iHUU. Although studies conducted before the wide adoption of perinatal ARV treatment reported heightened immune activation and/or exhaustion in iHEU[48,49], more recent studies have shown reduced or impaired memory T cells among iHEU compared to iHUU[17,50–52]. Such T cell alterations were associated with poor responses to vaccines and/or increased hospitalisation[50,51]. Other viral infections occurring during pregnancy such as SARS-Cov2 have also been documented to alter

infant immunity even in the absence of viral transmission[53,54]. Although these studies were limited in examining the difference in cord blood, early viral exposure resulted in lower frequencies of TNF-α and IFN-γ producing T cells[53]. Further longitudinal investigations are required to determine the impact of these early life cellular immune changes and their association to clinical outcomes later in life.

The differentiation of naïve T cells to different memory subsets are dependent on the TCR diversity of the naïve T cells in recognising the different pathogen derived antigens presented via major histo-compatibility complex (MHC) molecules on antigen presenting cells[10]. The poor T cell memory differentiation observed in iHEU could therefore be due to reduced pool of naïve TCR clones. This is supported by the observed differences in naïve TCRβ clonality between iHEU and iHUU being apparent from birth and correlated to the differentiated T cells preceding the period where impaired T cell memory differentiation was observed among iHEU.

Gabriel et al. reported lower TCRβ clonality in the cord blood of iHEU[42] implying clonal expansion occurring *in utero*, a phenomenon also observed in SARS-Cov2 exposed infants[54]. Our data further revealed that the memory T cells were the main contributors of reduced TCRβ diversity among iHEU compared to iHUU at birth. In addition to lower TCRβ diversity at birth, iHEU also had altered Vβ gene usage, with several genes being used more frequently relative to iHiUU. When using GLIPH2[37,38], however, we could not detect any predicted antigen specificity enriched among iHEU including HIV-specific clones. This contrasted with earlier studies showing higher frequencies of HIV-1 specific clones[42] and reactivity of T cells towards HIV proteins in iHEU[55]. What was evident instead was that naïve CD4+ and CD8+ TCRβ specificities in iHUU were enriched for clones targeting CMV, EBV and HCV antigens.

Since we obtained >95% purity after sorting for naïve and memory T cells, we regard this as minimal contamination of naïve T cells by memory T cells that could account for these enriched clones. It is possible that the abundance of naïve T cells compared to memory T cells in infant PBMC contributed to sampling bias resulting in enriched TCR clones amongst naïve T cells compared to the memory fraction. We also rationalised that since we did not include CD95 in our panel, we could not tease out stem cell-like memory T cells that would also express CD45RA and CCR7 similar to naïve cells and thus contribute to the observed outcome[56–58].

These scenarios, however, do not explain the lack of antigen specific TCR clonal enrichment among iHEU who exhibited skewed TCR clonality and Vβ gene usage. In addition, the length of CDR3 derived from these clones were typically short, a phenomenon common with neonatal TCR repertoire, and did not differ between iHEU and iHUU. Altogether, these observations possibly suggest non-antigen driven mechanisms that resulted in the loss of diversification of the iHEU TCR repertoire. Factors such as homeostatic proliferation under chronic maternal immune activation and proinflammatory milieu could give rise to virtual T cells[59–61], altered thymic selection since iHEU have been reported to have reduced thymic size relative to iHUU[43,62], or partially dysfunctional RAG system could contribute to iHEU having skewed TCR clonality. Since it is evident that reduced TCR diversity in iHEU is associated with impaired T cell memory differentiation and possibly increasing vulnerability to infection among iHEU[51], further investigations are required to decipher the mechanism responsible for the skewed TCR clonality in iHEU.

To better understand how these alterations in iHEU immunity could impact immunological responses we measured antibody response to common childhood vaccines such as to pertussis and rotavirus. Previous studies have consistently demonstrated that although maternally derived vaccine specific antibodies in iHEU are lower at birth, following vaccination the concentrations tend to increase to similar levels to those of iHUU, and for other vaccines, such as acellular pertussis vaccine and pneumococcal conjugate vaccine

(PCV), where iHEU antibody levels even surpass that of iHUU, albeit with poor functionality[63]. In our cohort we also observed a similar trend for anti-pertussis IgG levels being higher in iHEU compared to iHUU. Our data further revealed that among iHEU, vaccine-specific responses tend to have a high degree of variance compared to iHUU. This enabled us to classify infants either as "poor" or "good" vaccine responders, and therefore allowing us to show that, instead of T cells, the composition of NK cells pre-vaccination was superior in predicting vaccine-induced antibody responses post vaccination. This was consistent with other studies showing innate immune cells, including NK cells, being good baseline markers for predicting vaccine-induced antibody responses, albeit in adults[44,64]. NK cells either negatively or positively regulate adaptive immunity, including promoting immunoglobulin isotype switching and enhancing antibody production[65,66]. Here we show that the abundance of CD56loCD16loCD38+CD45RA+FcεRIγ+ NK cell subset prior to immunisation resulted in a "good" vaccine-induced antibody responses. This may be in agreement with Cuapio et al., who identified a CD56loNKG2C+ NK subset whose frequency at baseline correlated with anti-SARS-Cov2 antibody titres post-vaccination[64]. These data suggest: (a) that NK cell immunity may be important for vaccine-induced adaptive immunity and (b) induction of NK cells soon after birth could be an approach to enhance vaccine immunogenicity for childhood vaccines. Further mechanistic investigations are required to validate these observations.

The novelty of our study lies with the longitudinal follow-up of our infants and linking matched sample analysis, involving the ontogeny of T cells and NK cells from birth to 36 weeks of life and their relation to TCRβ, with vaccine-induced immunity. Limitations of our study included sample availability and characterising phenotypic changes of NK and T cells with inference to immune function by determining associations with TCR usage and vaccine-induced antibodies. More detailed cellular immune function would be required to validate the phenotypic characterisation. A further limitation was the inability to tease out the effect of HIV from ARV exposure since all pregnant women living with HIV receive ARV treatment as a standard of care[67].

In conclusion, our data show that immunity during infancy is impacted by HIV/ARV exposure in a sequential manner, starting with altered T cell clonality followed by delayed T cell memory differentiation. Of particular importance, is the finding that TCR clonotypic diversity is significantly lower in iHEU, with altered clonality of memory T cells soon after birth, suggesting *in utero* "priming" and the lower TCR diversity among naïve T cells associated with delayed T cell effector memory formation in these children. Although HIV/ARV exposure had a subtle impact on the NK cell trajectory, the relatively more abundant NK cell cluster in iHEU occurring of early in life was predictive of higher vaccine-induced antibody responses, suggesting the importance of innate immunity in early-life vaccine responses. Lastly, we show here a comprehensive phenotypic view of early life immune changes in response to HIV/ARV exposure, where these changes may well be linked to the observed vulnerability of co-morbidities in iHEU relative to iHUU.

## Methods
### Ethics statement
This research study complies with all relevant ethical regulations and was approved by the Human Research Ethics Committee (HREC) of University of Cape Town (HREC reference number 285/2012). All the women enrolled in the study provided signed informed consent for themselves and their respective infants.

### Study cohort
We analysed infants who were delivered by mothers living with and without HIV infection who were enrolled prospectively from birth and followed up until 9 months of age as previously described (Table S2)[68]. Briefly, pregnant women ≥18 years who recently delivered <12 h were

enrolled in the study together with their respective infants following signed informed consent. All mothers living with HIV received combined antiretroviral treatment during pregnancy under the Option B programme. The study enrolment was restricted to infants who were born by vaginal delivery and having birth weight ≥2.5 kg, gestation ≥36 weeks and no complication experienced during delivery. Of those delivered by mothers living with HIV, viral transmission was assessed by performing HIV DNA PCR test after 6 weeks of life, and those who tested positive for perinatal HIV infection were excluded from further analysis. The infants participating in this study received childhood vaccines according to the South African Extended Program of Immunisation (EPI). This included administration of acellular pertussis vaccine at weeks 6, 10 and 14 and rotavirus vaccine at weeks 6 and 14. Blood samples were collected at birth, weeks 4, 15 and 36 for isolation of plasma and peripheral blood mononuclear cells (PBMC). Demographic characteristics of the mothers and their respective infants are included in the study analysis and summarised in Table S2.

### Plasma and peripheral blood mononuclear cell processing

Infant blood samples (0.5−3 mL) were collected into sodium heparin tubes and processed within 6 h. Plasma and PBMC were isolated using ficoll centrifugation. Plasma samples were stored at −80 °C while PBMC were cryopreserved in 90% Fetal Calf serum (FCS) with 10% DMSO in liquid nitrogen.

### Intracellular and surface staining of PBMC for mass cytometry

A total of 278 infant samples (including 27 duplicates) were used to assess the longitudinal changes of immune cells using mass cytometry and TCR sequencing. PBMC samples were retrieved from liquid nitrogen and thawed at 37 °C before being transferred into RPMI 1640 media supplemented with 10% FCS and 10KU Benzonase. Cells were centrifuged, washed twice in PBS and counted using a TC20 cell counter (Biorad). We aimed to stain $2 \times 10^6$ viable cells for mass cytometry, however cell recovery varied by infant and age, with fewer cells collected at earlier time points due to smaller blood volumes. PBMC samples were split into 2 aliquots with $2 \times 10^6$ (or ¾ for cells with $<2 \times 10^6$) used for mass cytometry staining and $1 \times 10^6$ (or ¼ for cells with $<2 \times 10^6$) used for TCR sequencing. Lyophilised surface and intracellular antibody mixtures[69] stored at 4 °C were reconstituted in CyFACS buffer (PBS, 2% FCS) and permeabilization buffer (eBioscience permwash) respectively for cell labelling. Table S2 lists the antigen targets included in the mass cytometry antibody panel. Prior to antibody labelling, the cells were stained with cisplatin in PBS to determine cell viability followed by staining with surface antibodies for 20 min at room temperature. Cells were then fixed with 2% paraformaldehyde solution, permeabilised and stained with intracellular antibodies at 4 °C for 45 min. Upon which cells were washed with CyFACS buffer and resuspended with 2% paraformaldehyde solution containing Iridium DNA intercalator overnight at 4 °C. Samples were then washed with PBS and resuspended in MilliQ water containing EQ Four Element calibration beads (10% v,v, Fluidigm) prior to acquisition using CyTOF 2 instrument (DVS Sciences).

### Fluorescent activated cell sorting of naïve and memory T cells

The remaining PBMC from each sample were used to sort for naïve and memory CD4$^+$ and CD8$^+$ T cells using fluorescent activated cell sorting. Cells were labelled by surface staining using T cell markers including CD3, CD4 and CD8 and memory markers CD27, CD45RA and CCR7 (Table S2 and Fig. S3). Naïve cells were denoted as those co-expressing CCR7, CD45RA and CD27 while the other remaining cells were regarded as memory cells. BD FACSAria Fusion (BD) was used for 4-way sorting of CD4 and CD8 naïve and memory T cells from each sample and collected directly into FCS. Sorted cells were centrifuged, resuspended in cell RNAProtect (Qiagen) and stored at −40 °C until processing for TCR sequencing.

### Bulk TCR sequencing

RNA was purified from each sorted sample using the RNAeasy Plus Micro kit (Qiagen) and libraries were prepared for TCR sequencing as described by ref. 70 This was performed at the University of Cape Town. Briefly, purified RNA was reverse transcribed in a Rapid amplification of cDNA ends (RACE) reaction using SMARTScribe Reverse Transcriptase using previously designed oligos (isoC-5′- GTCAGAT GTGTATAAGAGACAGnnnnnnnnnnnCGATAGrGrGrG -3′-C3_Spacer and for the Poly A tail 5′- GTGTCACGTACAGAGTCATCttttttttttttttttt tttttttttttttt -3′ VN) that captures polyadenylated transcripts and introduces a 10 bp unique molecular identifier (UMI) at the 5′ end of the product. cDNA was purified with AmpureXP beads (Beckman Coulter) before whole transcriptome amplification using Advantage 2 Polymerase (Clontech) using oligos that introduce Illumina Nextera Multiplex Identifier (MID) P5 Adapter sequences. The libraries were then sent to Stanford University (Department of Microbiology and Immunology, Stanford University) in a blinded fashion, where TCRβ-specific amplification was achieved with Q5 Hot Start Master Mix (NEB) using constant region-specific oligos, simultaneously introducing P7 MID sequences. Final sequencing libraries were purified using SPRI-Select beads (Beckman Coulter), quantified using the Agilent TapeStation and pooled for sequencing. Paired-end sequencing was performed on an Illumina NovaSeq SP with 2 × 250 cycles, performed by the Chan-Zuckerberg Biohub Initiative.

### Quantification of vaccine antibody responses

Infant pertussis specific antibody responses were measured at birth, 4, 15 and 36 weeks of age using a commercial human IgG ELISA kit (Abcam). Plasma samples were diluted 1:100 into sample diluent and antibody titres were measured in duplicates according to manufacture instructions.

Rotavirus IgA titres were determined by enzyme immunoassay (EIA) and performed in a blinded manner in the Division of Infectious Disease, Department of Pediatrics, Cincinnati Childrens' Hospital Medical Centre, Cincinnati, Ohio. Briefly, purified rabbit anti-rotavirus IgG were immobilised on microtiter plate as capture antibodies. Lysates from rotavirus (strains RV3 and 8912) and mock infected cells (control) were added to the immobilised capture antibodies to bind rotavirus antigens and any uncaptured antigens were washed off with PBS, 0.05% Tween20. Reference standards, control and test samples were diluted in PBS, 0.05% Tween20, 1% non-fat dry milk and 50 uL added to microtiter plate for antigen binding. Bound antibodies were detected by biotyinylated goat anti-human IgA (Jackson Laboratories) and the addition of peroxidase conjugated avidin:biotin (Vector Laboratories, Inc., Burlingame, CA). Chromogenic signal was generated by addition of substrate O-phenylenediamine (Sigma) and reaction stopped after 30 min using 1 M H$_2$SO$_4$. Absorbance was measured at 492 $_{nm}$ on a Molecular Devices SpectraMax Plus plate reader. The reference standard was assigned 1000 arbitrary units (AU) and a four-parameter logistic regression was used to extrapolate anti-rotavirus IgA tires using SoftMax software.

Rotavirus neutralisation titres were determined as previously described[71]. In this study we used Wa G1P8, 1076 G4P6 and DS-1 G2P4 virus strains obtained from the National Institute of Health (NIH). Plasma samples were serially diluted and incubated with the rotavirus strain for neutralisation prior to adding the mixture to susceptible MA104 (monkey kidney) cell line for overnight incubation. Cell lysates were used to determine the level of un-neutralised rotavirus antigens using the EIA as described using guinea pig anti-rotavirus antiserum to measure captured rotavirus antigens. Rabbit anti-guinea pig IgG conjugated to horseradish peroxidase (Jackson ImmunoResearch) was used to detect bound antibodies, and chromogenic signal generated using OPD. The amount of rotavirus present in the resuspended lysate from each well was inversely related to the amount of neutralising antibody present in the plasma. Each plasma dilution series as

modelled using a logistic regression function. For each fitted curve the dilution which corresponds to a 40% response (ED$_{40}$), compared to the virus controls, was determined and reported as the neutralisation titer. The ED$_{40}$ represents the titre of the serum against a given virus, which represents a 60% reduction in amount of virus.

## Mass cytometry

**Processing.** Post CyTOF acquisition, the files were normalised using Premessa R package and target populations were manually gated using FlowJo software (version 10.5.3, TreeStar). The target populations with >1000 events were exported as FCS files for downstream analysis using R, an open-source statistical software[72]. All files for the target populations were transformed by applying inverse hyperbolic sine with a cofactor of 5 and subjected to doublet detection and removal using computeDoubletDensity from the scDblFinder package[73].

**Dimensional reduction.** Median marker expressions for each sample were used to compute Euclidean distance matrix to determine multidimensional scaling (MDS) coordinates using the cmdscale function in R. MDS analysis were used to visualise immune cell trajectories. Further, marker expression intensities were used to implement Uniform Manifold Approximation and Projection (UMAP) while preserving local density distribution for visualisation using densMAP function from the denvis R package[74].

**Cell Clustering.** High resolution clustering of the target cell populations was performed using FlowSOM algorithm and the resulting clusters grouped into metaclusters using ConsensusClusterPlus available in the CATALYST R package[75]. The immune cell clusters were visualised using hierarchical clustering and UMAP embedding. Centred log-ratios of the relative abundance of the cell clusters per sample were determined and used in Principal Component Analysis (PCA) for assessing immune cell compositional differences.

## TCR sequencing

**Pre-processing of TCR sequencing data.** Raw sequencing data was pre-processed by pRESTO[76]. Briefly, reads with mean Phred quality scores less than 20 were discarded, UMIs were extracted from the first 10 bp of read 1, and primer sequences masked. Next, a single consensus sequence was constructed from reads sharing the same UMI barcode and paired-end consensus sequences assembled into a full-length TCR sequence. In the case of non-overlapping mate-pairs, the human TRB reference was used to properly space non-overlapping reads. After removal of duplicate sequences, full-length reads were aligned and clonotypes assembled using MiXCR[77,78].

**T cell immune repertoire analysis.** To perform quality control on our bulk TCR sequencing data, we removed samples where fewer than 10 clones were detected (n = 44 samples) and where more clones were detected than cells were sorted (n = 18 samples). This yielded a total of 861 samples for downstream analysis. The immunarch package[79] operating in the open-source statistical software R was used for all downstream immune repertoire analysis, including calculation of CDR3β lengths and repertoire diversity by the Inverse Simpson index and richness using Choa1.

**Identification and annotation of T cell specificity groups.** The GLIPH2 algorithm was used to establish T cell specificity groups, clusters of CDR3 sequences that are predicted to bind to the same antigen, by discovering groups of CDR3β sequences that share either global or local motifs[37,38]. To assign specificity annotations to identified CDR3β sequence clusters, we adapted an approach described by ref. 80 Briefly, we collected human CDR3β sequences with known specificity from VDJdb (https://vdjdb.cdr3.net/) and identified CDR3β sequences within this database that could form TCR specificity groups.

From 47,107 CDR3β sequences, this process identified 11,648 specificity groups comprised of 25,759 unique CDR3β sequences. We next combined these 25,759 annotated sequences with 108,183 unique experimental CDR3β sequences from our bulk TCR repertoire profiling cohort. This was performed separately for CD4 and CD8 T, as there are differences between gene usage frequencies between CD4 and CD8 T cells that can impact specificity predictions. We prioritised TCR specificity groups with at least 3 distinct CDR3β sequences. This process yielded 17,154 CDR3β sequences in 11,629 specificity groups for CD4 T cells, and 15,107 CDR3β sequences in 9911 specificity groups for CD8 T cells.

## Statistical analysis

All statistical analysis and graphical visualisation of the data was performed on the open R software[72]. Spearman's rank correlation was used to test for the correlations between the frequencies of the cell clusters with either infant age or antibody responses and p values adjusted for multiple comparisons using false discovery rates (FDR). To compare differences in the abundance of cell clusters between iHEU and iHUU the generalised linear model from the diffcyt package was used[81]. Pairwise comparisons of medians between groups were performed using Wilcoxon-rank test or Kruskal-Wallis for multiple group comparisons and the p values adjusted for multiple comparisons using Benjamin Hochberg correction. To determine cell clusters that were predictive of antibody responses following vaccination, we used the multivariate modelling algorithm from MUVR package that incorporates recursive variable selection using repeated double cross-validation within partial least squares modelling[40].

## Reporting summary

Further information on research design is available in the Nature Portfolio Reporting Summary linked to this article.

# Data availability

FCS files for CyTOF data that was used for analysing infant immune trajectories and FACS data generated when sorting for naïve and memory CD4+ and CD8 + T cells are available at ImmPort repository (study accession SDY2463). FASTQ files generated from TCR RNA sequencing are available at Gene Expression Omnibus (accession GSE256410)The data files used for quantification of vaccine-specific antibody responses are available at Figshare repository. Source data are provided with this paper.

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

## Acknowledgements

We would like to acknowledge the technical staff that assisted in the completion of this study. This includes Euan Johnston, Emily Tangie and Carine Kilola who assisted with the CyTOF experiments, Mikayla Stabile, Michelle Leong, Kassandra Pinedo, and Nicole Prins, for assistance with TCR RNA-seq processing, Llyod Leach for assistance with measuring pertussis antibody responses and Monica McNeal for measuring rotavirus antibody responses. This work was supported by the Eunice Kennedy Shriver National Institute of Child Health and Human Development, National Institutes of Health R01HD102050 awarded to C.M.G. and H.B.J., and U01AI131302 to C.A.B., C.M.G. and H.B.J., and Fogarty International Training Center (D71TW012265) to C.M.G. and South African Medical Research Council (SA-MRC) to S.D. S.D. was supported by the Wellcome Trust International Training Fellowship (221995/Z/20/Z).

## Author contributions

Conceived by C.M.G., C.A.B., and H.B.J., Method set-up and validation by S.D., A.W., S.C., T.R., F.R., H.H., and M.D. Study participant enrolment, sample collection and processing by H.B.J. and B.A. Experimental investigations by S.D. Data and statistical analysis by S.D., A.W., and S.P.H. Original draft by S.D. Reviewed and edited by all authors.

## Competing interests

The authors declare no competing interests.
