## [Peer Review File · Nature Communications]

Premature skewing of T cell receptor clonality and delayed memory expansion in HIV-exposed infantsREVIEWER COMMENTS

Reviewer #1 (Remarks to the Author):

This is a longitudinal study to compare infants born to HIV positive mothers, but do not contract the virus, to infants born to HIV negative mothers. The longitudinal aspect of the study makes it noteworthy as collecting samples from infants is very difficult. Here, Dzanibe and colleagues sought to determine developmental differences between those infants who were HIV exposed and those who were not on infant immune ontogeny. They use mass cytometry to describe differences in NK cell populations. TCR analysis was completed as well and demonstrates decreased diversity in ν beta clonotypic diversity.

I found the paper extremely difficult to follow. Ultimately, it is descriptive. Because of the nature of the work and the limited sample amount collected from a baby, it is hard to do mechanistic studies. However, there is no clear theme to the description of the evolution of the infant immune system. Moreover, the authors have chosen to focus their description on the difference between HIV exposed and non-exposed babies, when there are actually very few differences. The differences that they do cite are inconsistent and it's not clear what the meaning of a particular difference is. The title of the paper indicates that there is: 1. disrupted memory T cell expansion and 2. Premature skewing of T cell clonality.

First, the premise that there is disrupted CD8+ memory T cell expansion is based on there being lower mean frequencies of clusters 2 (NKT cells), 3 (naïve like activated cells) and 7 (proliferating EM). By week 36, it is only clusters 2 and 3 which are different (lines 211-214). For CD4 T cells, 4,5,6,7,9 and 11 were lower in the exposed infants at 15 weeks and clusters 5,7,9 remained low at 36 weeks. I could not follow how these results indicated that memory T cell expansion was disrupted, particularly because the couple of differences which were present at 15 weeks disappeared at 36 weeks.

Second, the authors show some TCR analysis which demonstrates differences in diversity and richness in the TCR repertoire. The TCR analysis is cursory though as it only examined the diversity and richness score. There is no information about CDR3 length and n1 and n2 insertions to demonstrate whether the TCR repertoire is more germline. I suspect that the decrease in diversity is not because of antigen exposure, but rather there is a stunting of the T cell receptor diversification. The authors fail to acknowledge this in the discussion and fail to cite several important papers on the development of the infant TCR repertoire. It is also confusing that the unexposed infants had greater expansion of specific clonotypes against known antigens. Enrichment of particular clones should decrease diversity and richness. To have decreased diversity in the exposed infants, there needs to be expansion of clones. So what clones are these? They begin to address this question in the discussion, line 423. They propose that they are virtual T cells, a product of homeostatic proliferation. Perhaps, but there is no connection to infant immune ontogeny. Again, there is a missed opportunity to cite the appropriate literature. They do some functional analysis by investigating antibody responses to two vaccines. However, when they build their predictive model, they lump the exposed and unexposed together, which further supports the fact that the difference between the exposed and unexposed infants is slight. Ultimately, there are strong developmental forces at work which dictate infant immune ontogeny.

Reviewer #2 (Remarks to the Author):

The novelty of this study lies with the longitudinal follow-up of infants from birth to 36 weeks of life with matched samples for multiple analyses to detail the ontogeny of T and NK cells and their relationship with vaccination success. There is interest in the effect of in utero exposure to HIV (+/- ARV which couldn't be assessed) interrogated here. This paper used mass cytometry to look at immune cell differences and the later responses to vaccinations and investigated TCR clonality over time.

They presented some evidence that HIV/ARV exposure disrupts innate and adaptive immunity but a lot was overstated. The finding that TCR clonotypic diversity is significantly lower in iHEU from birth suggests in utero skewing well before T cell memory formation. But, as noted by the authors others have already reported lower β TCR clonality in the cord blood of iHEU

Fig 1 is pretty superfluous as several groups have published on the immune trajectory post birth so this data is not novel. There have been groups looking at immune functions post SARS-CoV-2 exposure which were not cited.

One of the main flaws is there was actually no significant differences in the NK cell groups between HEU and HUU (Suppl 2C) so really the reference to this in the text (line 229-234) should be removed. All of these points are not significantly different after adjustment-which is rightly used in these studies...and even when looking at the PCA there was only one significant difference at week 4-when all other time points were equivalent. Hence, the data doesn't really support the conclusions 'NK cell phenotype and cluster composition differed between iHEU and iHUU at birth and week 4'-this wasn't significant at birth yet this is implied in this sentence.

Although significant differences in clusters between HEU and HUU as a group were seen in CD4 T cells for example (Fig 2), the heat plots demonstrate many HEU samples (nearly half in some clusters) did still look similar to HUU-what is the difference between these different HEU samples? Unbiased clustering may put those samples into different groups, some more like HUU and some not? Are there any differences between the infants within the HEU that may explain this and that you could comment on?

Why do the heat plots for CD8 T cells in Fig 2 have less samples in them than CD4? They should be the same? Where is the rest of the data for CD8 from the samples that CD4 was measured in? How many memory cells were sorted from the HEU/HUU from which the TCR repertoire was assessed-this could skew the data as the memory cell numbers are low? Similarly, it is not clear how many individual samples were analysed for the TCR repertoire-you state 934 sorted cell fractions were assessed which is not divisible by 4 implying you didn't sort 4 fractions from every sample? This all needs to be clearer and more transparent. The TCR repertoire and naïve T cell cluster correlations that you make-were these done on the total cell cluster analysis or only using samples from which the TCR repertoire was also assessed? Were cell correlations with TCRb diversity only seen in HEU and not in HUU at birth? It isn't clear if that correlation was assessed and was not apparent?

Whilst we may assume that the infants in this study were born by vaginal delivery and not caesarean section, this is not explicitly stated and as delivery mode is a key variable in ongoing immune function this needs to be clarified. Also, there was a significant difference in age of mothers between the two groups but this was not discussed.

There is a lot of data regarding the role of Tregs post birth but no Tregs identified in the analysis? It seems that FOXP3 was not in the Ab panel, this needs to be discussed as a limitation

Minor corrections needed:

There were a lot of typos/missing information. Please correct

INF γ lines 86/87/88 . Be consistent IFN-g or INF-g

Line 157 parsed manually gated CD4 and CD8 cells (presumably paired?)

Unclear why 'expanded NK cell subsets in iHEU' would be detrimental and not a positive-please expand or remove

Not sure Per is a well known abbreviation for perforin-change and also be consistent, sometimes this is labelled as Per, other times perforin

Table S1/S2 are in the wrong order as S2 discussed first in the manuscript?

Please number the clusters in Fig S2 so we can see how they relate to the clusters in Fig 2-not just colour coded

Cluster 5 in CD8, what other markers are on this cluster -this is distinguished from cluster 1 by presence of CD45RA, CD27?

X axis labels on Fig 4a are absent?

Fig 4c/d $p=0.05/p=0.02$ have same size circles in the key.. remove $p=0.75$ info from the dataset as ns and I have no clue what the tiny circles represent as no key for those at all!

Did the authors assess any other populations to see if they related to subsequent Ab responses -B cells, APCs? Did you only put the T/Nk data into the prediction model?

Whilst the authors suggested a functional link between NK cell and Ab responses, more discussion about the ACTUAL NK populations that were linked compared to others that were not-what do those specific populations suggest? If you did the prediction model between responders and non-responders but further separated based on HEU/HUU what happens? Do any of the correlations change if you separate based on HIV exposure?

Why are responses to one vaccination greater in HEU than HUU but vice versa for a different vaccination? Some discussion should be made of this

Reviewer #3 (Remarks to the Author):

Review – NCOMMS-23-23072A-Z

The implementation improved and safe antiretroviral therapy options has resulted in a drastic reduction of in utero infection of infants with HIV. However, the concurrent increase in HIV-exposed uninfected infants presents as another serious health problem, as many iHEU experience increased susceptibility to infections. The study by Dzanibe et al. aims to identify underlying immune differences between iHEU and HIV unexposed and uninfected infants.

Performing mass spectrometry, the investigators identified defined the immune trajectory of CD4+ and CD8+ T cells, as well as NK cells in the first 9 months of life. The results demonstrated differences in the differentiation of specific T cell clusters between iHEU and iHUU, especially at early time points. In addition, memory T cell populations of iHEU exhibited reduced TCR beta clonotypic diversity compared to iHUU. The immune analysis further revealed divergence between NK cell populations depending on the infant's HIV exposure. Independent of HIV exposure, the presence of specific NK cell populations at birth was associated with vaccine responses to rotavirus vaccine and acellular pertussis vaccine at 9 months of age.

Together, the presented findings add to our knowledge of immune development in African infant populations and provide new insights into kinetic and qualitative differences of immune ontogeny between iHEU and iHUU. As such, the study results are significant.

The clarity of the manuscript could be improved, and the following comments should be addressed:

1. The authors talk about a longitudinal study and in the Discussion (lines 461ff) they refer to the "...novelty of our study lies with the longitudinal follow-up of our infants with matched samples...". Please clarify, if this statement applies only to the infant samples from weeks 4, 15, and 36 (n=53, n=52, and n=53, respectively). Do the samples of cord blood (n=5) and 4-week-old infants (n=44) represent different groups of infants?
2. How were matched samples versus cross-sectional samples considered in the various analyses, and in particular in the immune trajectory analysis? Considering the high variability in data at each time point, would a separate analysis using only matched samples provide more concise data?
3. The immune panel was clearly biased towards T cell and NK cell populations. Monocyte, dendritic cell, and B cell differentiation was not tested. It might be possible to include monocytes with characterization by CD14 and CD16, this might be interesting for iHEU? This should be stated in the methods. Please also provide a rationale.
4. Considering the association between NK cell populations at birth and vaccine responses at wk 36, one wonders if there would have been correlations between antigen-presenting cell populations and vaccine responses, too, and whether B cell clusters and B cell differentiation would correlate with vaccine responses. The lack of any associations between T cell populations and vaccine responses could also be explained by the lack of consideration of follicular T helper cell responses in the current study.
5. The manuscript title implies that this study was focused on differences between iHEU and iHUU in T cell development. Yet, even in the abstract, the authors point out the differences in NK cell development and point to the important finding that birth NK cell clusters correlate with vaccine responses. The latter correlation, however, is independent of HIV exposure. Therefore, the reviewer wonders whether different sets of interesting data were combined in this manuscript. The link between the data is not always clear and sometimes almost artificial. Maybe, data of general immune development for T cells and NK cells should precede the description of developmental differences between iHEU and iHUU? For lack of functional studies, one could then envision examining TCR diversity and vaccine-induced antibody responses.
6. The finding that iHUU have a preference for HIV among the TCR beta clones is curious and worth some discussion.
7. On Page 10, line 229, should "iHEU" read "iHUU had higher frequencies..."?
8. Please provide a reference for your statement that immune populations at week 36 had an adult-like phenotype (page 7, line 151).

We thank you for the opportunity to submit a revised version of the above-referenced manuscript. Below is a point-by-point response to the reviewers' comments and suggestions.

Reviewer #1

This is a longitudinal study to compare infants born to HIV positive mothers, but do not contract the virus, to infants born to HIV negative mothers. The longitudinal aspect of the study makes it noteworthy as collecting samples from infants is very difficult. Here, Dzanibe and colleagues sought to determine developmental differences between those infants who were HIV exposed and those who were not on infant immune ontogeny. They use mass cytometry to describe differences in NK cell populations. TCR analysis was completed as well and demonstrates decreased diversity in v beta clonotypic diversity.

I found the paper extremely difficult to follow. Ultimately, it is descriptive. Because of the nature of the work and the limited sample amount collected from a baby, it is hard to do mechanistic studies. However, there is no clear theme to the description of the evolution of the infant immune system. Moreover, the authors have chosen to focus their description on the difference between HIV exposed and non-exposed babies, when there are actually very few differences. The differences that they do cite are inconsistent and it's not clear what the meaning of a particular difference is. The title of the paper indicates that there is: 1. disrupted memory T cell expansion and 2. Premature skewing of T cell clonality.

First, the premise that there is disrupted CD8+ memory T cell expansion is based on there being lower mean frequencies of clusters 2 (NKT cells), 3 (naïve like activated cells) and 7 (proliferating EM). By week 36, it is only clusters 2 and 3 which are different (lines 211-214). For CD4 T cells, 4,5,6,7,9 and 11 were lower in the exposed infants at 15 weeks and clusters 5,7,9 remained low at 36 weeks. I could not follow how these results indicated that memory T cell expansion was disrupted, particularly because the couple of differences which were present at 15 weeks disappeared at 36 weeks.

Second, the authors show some TCR analysis which demonstrates differences in diversity and richness in the TCR repertoire. The TCR analysis is cursory though as it only examined the diversity and richness score. There is no information about CDR3 length and n1 and n2

insertions to demonstrate whether the TCR repertoire is more germline. I suspect that the decrease in diversity is not because of antigen exposure, but rather there is a stunting of the T cell receptor diversification. The authors fail to acknowledge this in the discussion and fail to cite several important papers on the development of the infant TCR repertoire. It is also confusing that the unexposed infants had greater expansion of specific clonotypes against known antigens. Enrichment of particular clones should decrease diversity and richness. To have decreased diversity in the exposed infants, there needs to be expansion of clones. So what clones are these? They begin to address this question in the discussion, line 423. They propose that they are virtual T cells, a product of homeostatic proliferation. Perhaps, but there is no connection to infant immune ontogeny. Again, there is a missed opportunity to cite the appropriate literature.

They do some functional analysis by investigating antibody responses to two vaccines. However, when they build their predictive model, they lump the exposed and unexposed together, which further supports the fact that the difference between the exposed and unexposed infants is slight. Ultimately, there are strong developmental forces at work which dictate infant immune ontogeny.

Rebuttal

I found the paper extremely difficult to follow. Ultimately, it is descriptive. Because of the nature of the work and the limited sample amount collected from a baby, it is hard to do mechanistic studies. However, there is no clear theme to the description of the evolution of the infant immune system.

We have streamlined the paper and re-ordered the data-sets to make it easier to follow. We also acknowledge that this study was limited to only describing the phenotypic changes of the infant immune system adjunct to the TCR repertoire. Due to limited PBMC samples collected from the infant it was not feasible to include other related mechanistic investigations and thus we opted to measure differences in plasma antibody responses to childhood vaccines. These limitations are address in the discussion section.

Extract **Line 485-489** “Limitations of our study included sample availability and characterizing phenotypic changes of NK and T cells with inference to immune function by determining associations with TCR usage and vaccine-induced antibodies. More detailed cellular immune function would be required to validate the phenotypic characterization.”

We have also revised and restructured the results section where we initially discuss the longitudinal development of the overall infant immune system and the lineage immune trajectory corroborating earlier studies and thus validating our analytical approach.

Lines 114-139: “Immune cell transition from birth to 9-months of age” and **Lines 141-187:** “Distinct temporal maturation trajectories of NK cells stabilizing at 4 months coinciding with rapid CD8⁺ T cell divergence.”

Following showing the typical maturation immune trajectory in infants, we demonstrate how specific immune cell subsets are altered by maternal HIV infection.

Lines 188-221: “Memory maturation of CD4 and CD8 T cells is disrupted by HIV exposure after three months of life.”

Moreover, the authors have chosen to focus their description on the difference between HIV exposed and non-exposed babies, when there are actually very few differences. The differences that they do cite are inconsistent and it's not clear what the meaning of a particular difference is.

We agree that some of the differences were perhaps nuanced. We have since rewritten aspects of the paper to focus on the major differences, being the impact of maternal HIV infection on early life TCR skewing, delayed memory T cell maturation and how an NK cell population in the first 4 weeks of life predicts vaccine-induced antibody responses later in life. We have also revised the manuscript to highlight the connection between the differences observed in the memory T cells and how this associate with skewing of the TCR repertoire.

The following are modified descriptions from the revised paper that show these changes:

Line 213-218: Describing differences in NK cells at week 4 between iHEU and iHUU.

“The composition of NK cells differed by HIV-exposure only at week 4 (Figure 2A), partly driven by higher frequencies of the differentiated (CD56^{lo}NKG2A⁻CD57^{lo/+}) NK cell cluster 1 (CD56^{lo}CD16^{lo}Perforin⁺CD38⁺CD45RA⁺FcER1y⁺, p=0.04, p.adj=0.2) and cluster 5 (CD56^{lo}CD16⁺Perforin⁺CD57⁺CD45RA⁺CD38⁺ (p=0.03, p.adj=0.2) in iHEU compared to iHUU (Figure S4A), although these difference were not significant after correcting for multiple testing.”

Line 197-204: iHEU at weeks 15 and 36 had lower frequencies of memory differentiated CD4⁺ T cells compared to iHUU.

“Differential abundance testing using generalized mixed model revealed that the divergent CD4⁺ T cell phenotypes in iHEU were due to lower frequencies of closely related activated and proliferating (HLA-DR⁺Ki67⁺) clusters 4 & 11, and memory differentiated cytolytic (PD-1⁺CD57⁺Perforin⁺)^{31,32} clusters 5- 9 compared to iHUU at week 15, with the differentiated cytolytic clusters 5, 7 and 9 remaining significantly lower in iHEU at week 36 (Figure 2D, 2E & S3B). The CD4⁺ Th2-like cluster 1 (CCR4⁺CD27⁺CD127⁺) was, however, significantly higher in iHEU compared to iHUU at week 15 although by week 36 was no longer significantly elevated.”

Line 205-211: iHEU at week 36 had lower frequencies of differentiated CD8⁺ T cells.

“For CD8⁺ T cells, the compositional differences were partly driven by lower mean frequencies for clusters 2 (NKT-like cells: CD56⁺CD16⁺NKp30⁺NKp46⁺NKG2A⁺Perforin⁺)³³, 3 (Naïve-like activated cell: CD45RA⁺CD27⁺CCR7⁺CXCR3⁺CCR4⁺HLA-DR⁺) and 7 (proliferating EM: Ki67⁺CD45RA⁺CD27⁺CCR7⁻Perforin⁺) in iHEU compared to iHUU at week 15 (Figure 2I). By week 36, only the mean frequencies of clusters 2 (NKT-like cells) and 3 (Naïve-like cells) remained significantly lower in iHEU compared to iHUU (Figure 2F & 2G).

Line 254-256: iHEU had decreased naïve TCRβ clones later in life

“...within the naïve T cell compartment; TCRβ diversity was significantly lower in iHEU compared to iHUU at week 15 and 36 for CD4⁺ T cells, and at birth and week 36 for CD8⁺ T cells (Figure 3A).”

Line 276-287: The naïve TCRβ diversity was correlated to immune T cells that were negatively impacted by HIV-exposure.

“...the TCRβ diversity scores derived from sorted naïve T cells in iHEU were positively correlated to the frequencies of CD4⁺ T cell clusters (4-6, 9-11) and CD8⁺ T cell clusters (2,3,7) (Figure 3C). It is noteworthy that the observed T cell clusters that were correlated to the naïve TCRβ diversity included the HLA-DR⁺Ki67⁺CD4⁺ T cell clusters (4 & 11), memory differentiated CD4⁺ T cell clusters (5,6, 9 & 11) and the CD8⁺ T cell clusters (2,3 & 7) that were significantly lower amongst iHEU at week 15 and 36 (Figure 2D-G). Relative to the naïve CD4⁺ and CD8⁺ T cell clusters, the T cell clusters that were correlated with TCRβ displayed varying

expression levels of PD-1 suggesting TCR activation (Figure S3B & S3C)^{34,35}. For the proliferating activated CD4⁺ T cells (cluster 4 & 11) and the terminally differentiated cytolytic CD4⁺ T cells (cluster 5 & 6), these correlations were evident from birth up until week 15.”

Line 394-403: Discussion

“HIV exposure altered the immunological compositions of NK cells and T cells at different stages during infancy, with the T cell memory maturation being significantly impacted later in life. The impaired T cell memory maturation among iHEU could be in part be due to diminished naïve TCR clonal pools observed in iHEU relative to iHUU. The skewing of TCR clonality most likely initiating *in utero* as suggest by difference in infant groups observed as early as birth.^{41,42} We further show that pre-vaccine NK cell composition can predict vaccine-induced antibody levels in infants better than T cells and is consistent with studies reporting baseline immunological signatures that are predictive of vaccine responses.⁴³”

Line 416-423: Discussion

“The differentiation of naïve T cells to different memory subsets are dependent on the TCR diversity of the naïve T cells in recognizing the different pathogen derived antigens presented via major histocompatibility complex (MHC) molecules on antigen presenting cells.⁵⁵ The poor T cell memory differentiation observed in iHEU could therefore be due to reduced pool of naïve TCR clones. This is supported by the observed differences in naïve TCR β clonality between iHEU and iHUU being apparent from birth and correlated to the differentiated T cells preceding the period where impaired T cell memory differentiation was observed among iHEU.”

The title of the paper indicates that there is: 1. disrupted memory T cell expansion and 2. Premature skewing of T cell clonality. First, the premise that there is disrupted CD8+ memory T cell expansion is based on there being lower mean frequencies of clusters 2 (NKT cells), 3 (naïve like activated cells) and 7 (proliferating EM). By week 36, it is only clusters 2 and 3 which are different (lines 211-214). For CD4 T cells, 4,5,6,7,9 and 11 were lower in the exposed infants at 15 weeks and clusters 5,7,9 remained low at 36 weeks. I could not follow how these results indicated that memory T cell expansion was disrupted, particularly because the couple of differences which were present at 15 weeks disappeared at 36 weeks.

We have replaced the word disrupted with “Delayed” in the title. This more aptly describes the data where memory populations were only observed to be significantly lower in iHEU from week 15.

The differences between iHEU and iHUU are highlighted in the previous point above demonstrating how altered memory differentiation in iHEU coincided with lower TCR diversity. These outcomes suggest a possible mechanism for the observed outcomes in iHEU although further investigations are required for validation.

Second, the authors show some TCR analysis which demonstrates differences in diversity and richness in the TCR repertoire. The TCR analysis is cursory though as it only examined the diversity and richness score. There is no information about CDR3 length and n1 and n2 insertions to demonstrate whether the TCR repertoire is more germline. I suspect that the decrease in diversity is not because of antigen exposure, but rather there is a stunting of the T cell receptor diversification. The authors fail to acknowledge this in the discussion and fail to cite several important papers on the development of the infant TCR repertoire.

This is a good point. We show in Figure S5C CDR3 length distribution, where we did not observe differences in the CDR3 length distribution by infant age or HIV-exposure status. It is possible that the reduced TCR diversification could be due to other factor rather than the expansion of clones with specificity to a select antigens. This is supported by evidence from our GLIPH2 analysis (Figure 3D) that shows enrichment of clones in iHUU rather than iHEU.

Line 239-240:

“CDR3 lengths of the clonotypes were evenly distributed across infant age and between iHEU and iHUU (Figure S3C).”

Line 298-300:

“TCR specificities from sorted naïve CD4⁺ T cells in iHUU appeared enriched with specificities for cytomegalovirus (CMV), Hepatitis C virus (HCV), SARS-Cov2 and HIV-1 (Figure 3C).”

Discussion Line 427-433:

“In addition to lower TCR diversity at birth, iHEU also had altered V β gene usage, with several genes being used more frequently relative to iHiUU. When using GLIPH2^{36,37}, however, we could not detect any predicted antigen specificity enriched among iHEU including HIV-specific clones. This contrasted with earlier studies showing higher frequencies of HIV-1 specific clones⁴¹ and reactivity of T cells towards HIV proteins in iHEU⁵⁶. What was evident instead

was that naïve CD4⁺ and CD8⁺ TCRβ specificities in iHUU were enriched for clones targeting CMV, EBV and HCV antigens.”

Discussion **Line 443-452:**

“In addition, the length of CDR3 derived from these clones were typically short, a phenomenon common with neonatal TCR repertoire, and did not differ between iHEU and iHUU. Altogether, these observations possibly suggest non-antigen driven mechanisms that resulted in the loss of diversification of the iHEU TCR repertoire. Factors such as homeostatic proliferation under chronic maternal immune activation and proinflammatory milieu could give rise to virtual T cells,^{60–62} altered thymic selection since iHEU have been reported to have reduced thymic size relative to iHUU,^{42,63} or partially dysfunctional RAG system could contribute to iHEU having skewed TCR clonality.”

It is also confusing that the unexposed infants had greater expansion of specific clonotypes against known antigens. Enrichment of particular clones should decrease diversity and richness. To have decreased diversity in the exposed infants, there needs to be expansion of clones. So what clones are these? They begin to address this question in the discussion, line 423. They propose that they are virtual T cells, a product of homeostatic proliferation. Perhaps, but there is no connection to infant immune ontogeny. Again, there is a missed opportunity to cite the appropriate literature.

This also intrigued us and we regard this as one of the interesting points of the paper. To try and make this less confusing, we have revised the manuscript to discuss the possible permutation that would contribute to such outcomes (highlighted in yellow).

Line 424-455:

“Gabriel *et al.* reported lower TCRβ clonality in the cord blood of iHEU,⁴¹ implying clonal expansion occurring *in utero*. Our data further revealed that memory T cells were the main contributors of reduced TCRβ diversity among iHEU compared to iHUU at birth. In addition to lower TCR diversity at birth, iHEU also had altered Vβ gene usage, with several genes being used more frequently relative to iHiUU. When using GLIPH2^{36,37}, however, we could not detect any predicted antigen specificity enriched among iHEU including HIV-specific clones. This contrasted with earlier studies showing higher frequencies of HIV-1 specific clones⁴¹ and reactivity of T cells towards HIV proteins in iHEU⁵⁶. What was evident instead was that naïve

CD4⁺ and CD8⁺ TCR β specificities in iHUU were enriched for clones targeting CMV, EBV and HCV antigens.

Since we obtained >95% purity after sorting for naïve and memory T cells, we regard this as minimal contamination of naïve T cells by memory T cells that could account for these enriched clones. It is possible that the abundance of naïve T cells compared to memory T cells in infant PBMC contributed to sampling bias resulting in enriched TCR clones amongst naïve T cells compared to the memory fraction. We also rationalized that since we did not include CD95 in our panel, we could not tease out stem cell-like memory T cells that would also express CD45RA and CCR7 similar to naïve cells and thus contribute to the observed outcome^{57–59}.

These scenarios, however, do not explain the lack of antigen specific TCR clonal enrichment among iHEU who exhibited skewed TCR clonality and V β gene usage. In addition, the length of CDR3 derived from these clones were typically short, a phenomenon common with neonatal TCR repertoire, and did not differ between iHEU and iHUU. Altogether, these observed outcomes suggest other non-antigen driven mechanisms that resulted in the loss of diversification of the iHEU TCR repertoire. Factors such as homeostatic proliferation under chronic maternal immune activation and proinflammatory milieu could give rise to virtual T cells,^{60–62} altered thymic selection since iHEU have been reported to have reduced thymic size relative to iHUU,^{42,63} or partially dysfunctional RAG system could contribute to iHEU having skewed TCR clonality. Since it is evident that reduced TCR diversity in iHEU is associated with impaired T cell memory differentiation and possibly increasing vulnerability to infection among iHEU,⁵³ further investigations are required to decipher the mechanism responsible for the skewed TCR clonality in iHEU.”

They do some functional analysis by investigating antibody responses to two vaccines. However, when they build their predictive model, they lump the exposed and unexposed together, which further supports the fact that the difference between the exposed and unexposed infants is slight. Ultimately, there are strong developmental forces at work which dictate infant immune ontogeny.

The decision to combine both iHEU and iHUU was due to the limited sample size for infants having both antibody responses measured pre-and post-vaccination and the availability of matched mass cytometry data. We do acknowledge that these predictive models require

further validation in a larger infant population and consider these results as preliminary indicators of baseline immunological composition predictive of vaccine response later in life.

Reviewer #2:

The novelty of this study lies with the longitudinal follow-up of infants from birth to 36 weeks of life with matched samples for multiple analyses to detail the ontogeny of T and NK cells and their relationship with vaccination success. There is interest in the effect of in utero exposure to HIV (+/- ARV which couldn't be assessed) interrogated here. This paper used mass cytometry to look at immune cell differences and the later responses to vaccinations and investigated TCR clonality over time.

They presented some evidence that HIV/ARV exposure disrupts innate and adaptive immunity but a lot was overstated. The finding that TCR clonotypic diversity is significantly lower in iHEU from birth suggests in utero skewing well before T cell memory formation. But, as noted by the authors others have already reported lower β TCR clonality in the cord blood of iHEU Fig 1 is pretty superfluous as several groups have published on the immune trajectory post birth so this data is not novel. There have been groups looking at immune functions post SARSCoV-2 exposure which were not cited.

One of the main flaws is there was actually no significant differences in the NK cell groups between HEU and HUU (Suppl 2C) so really the reference to this in the text (line 229-234) should be removed. All of these points are not significantly different after adjustment-which is rightly used in these studies...and even when looking at the PCA there was only one significant difference at week 4-when all other time points were equivalent. Hence, the data doesn't really support the conclusions 'NK cell phenotype and cluster composition differed between iHEU and iHUU at birth and week 4'-this wasn't significant at birth yet this is implied in this sentence. Although significant differences in clusters between HEU and HUU as a group were seen in CD4 T cells for example (Fig 2), the heat plots demonstrate many HEU samples (nearly half in some clusters) did still look similar to HUU-what is the difference between these different HEU samples? Unbiased clustering may put those samples into different groups, some more like HUU and some not? Are there any differences between the infants within the HEU that may explain this and that you could comment on?

Why do the heat plots for CD8 T cells in Fig 2 have less samples in them than CD4? They should be the same? Where is the rest of the data for CD8 from the samples that CD4 was measured in?

How many memory cells were sorted from the HEU/HUU from which the TCR repertoire was assessed-this could skew the data as the memory cell numbers are low? Similarly, it is not clear how many individual samples were analysed for the TCR repertoire-you state 934 sorted cell fractions were assessed which is not divisible by 4 implying you didn't sort 4 fractions from

every sample? This all needs to be clearer and more transparent. The TCR repertoire and naïve T cell cluster correlations that you make-were these done on the total cell cluster analysis or only using samples from which the TCR repertoire was also assessed? Were cell correlations with TCRb diversity only seen in HEU and not in HUU at birth? It isn't clear if that correlation was assessed and was not apparent?

Whilst we may assume that the infants in this study were born by vaginal delivery and not caesarean section, this is not explicitly stated and as delivery mode is a key variable in ongoing immune function this needs to be clarified. Also, there was a significant difference in age of mothers between the two groups but this was not discussed.

There is a lot of data regarding the role of Tregs post birth but no Tregs identified in the analysis? It seems that FOXP3 was not in the Ab panel, this needs to be discussed as a limitation

Minor corrections needed:

There were a lot of typos/missing information. Please correct

INF γ lines 86/87/88 . Be consistent IFN-g or INF-g

Line 157 parsed manually gated CD4 and CD8 cells (presumably paired?)

Unclear why 'expanded NK cell subsets in iHEU' would be detrimental and not a positive- please expand or remove

Not sure Per is a well known abbreviation for perforin-change and also be consistent, sometimes this is labelled as Per, other times perforin

Table S1/S2 are in the wrong order as S2 discussed first in the manuscript?

Please number the clusters in Fig S2 so we can see how they relate to the clusters in Fig 2-not just colour coded

Cluster 5 in CD8, what other markers are on this cluster -this is distinguished from cluster 1 by presence of CD45RA, CD27?

X axis labels on Fig 4a are absent?

Fig 4c/d p=0.05/p=0.02 have same size circles in the key.. remove p=0.75 info from the dataset as ns and I have no clue what the tiny circles represent as no key for those at all!

Did the authors assess any other populations to see if they related to subsequent Ab responses -B cells, APCs? Did you only put the T/Nk data into the prediction model?

Whilst the authors suggested a functional link between NK cell and Ab responses, more discussion about the ACTUAL NK populations that were linked compared to others that were not-what do those specific populations suggest? If you did the prediction model between responders and non-responders but further separated based on HEU/HUU what happens? Do

any of the correlations change if you separate based on HIV exposure?

Why are responses to one vaccination greater in HEU than HUU but vice versa for a different vaccination? Some discussion should be made of this

Rebuttal:

The finding that TCR clonotypic diversity is significantly lower in iHEU from birth suggests in utero skewing well before T cell memory formation. But, as noted by the authors others have already reported lower β TCR clonality in the cord blood of iHEU.

We acknowledge that lower TCR β clonal diversity has been reported in cord blood of iHEU compared to iHUU, but our study significantly differs in two ways: a) that the lower TCR β diversity at birth is primarily compartmentalized to the memory T cells and b) differences in TCR β clonality between iHEU and iHUU occur beyond birth and into the perinatal and infant period.

Results Line 247-256:

“...iHEU memory TCR β clonotypes had relatively lower diversity compared to iHUU at birth, (Figure 3A). Similarly, richness (the number of unique TCR β clones) was significantly lower in iHEU relative to iHUU for both memory CD4⁺ and CD8⁺ T cells at birth (Figure 3B). This remained statistically significant for memory CD4⁺ T cells at week 4 (Figure 3B).

In contrast to TCR β of memory T cells, in which significant differences were found early life, within the naïve T cell compartment; TCR β diversity was significantly lower in iHEU compared to iHUU at week 15 and 36 for CD4⁺ T cells, and at birth and week 36 for CD8⁺ T cells (Figure 3A).”

An interesting finding was that differences in TCR β clonality between iHEU and iHUU later in life were observed within the naïve T cell pool. When TCR β diversity was correlated to T cell subsets/clusters identified by mass cytometry, differentiated memory T cells that were significantly lower in iHEU compared to iHUU were positively correlated to TCR β diversity score for iHEU. This is reflected in the results and further elaborated in the discussion.

Results Line 276-282:

“...the TCR β diversity scores derived from sorted naïve T cells in iHEU were positively correlated to the frequencies of CD4⁺ T cell clusters (4-6, 9-11) and CD8⁺ T cell clusters (2,3,7) (Figure 3C). It is noteworthy that the observed T cell clusters that were correlated to the naïve TCR β diversity included the HLA-DR⁺Ki67⁺CD4⁺ T cell clusters (4 & 11), memory differentiated CD4⁺ T cell clusters (5,6, 9 & 11) and the CD8⁺ T cell clusters (2,3 & 7) that were significantly lower amongst iHEU at week 15 and 36 (Figure 2D-G).”

Discussion **Line 396-403:**

“The impaired T cell memory maturation among iHEU could be in part be due to diminished naïve TCR clonal pools observed in iHEU relative to iHUU. The skewing of TCR clonality most likely initiating *in utero* as suggest by difference in infant groups observed as early as birth.^{41,42} We further show that pre-vaccine NK cell composition can predict vaccine-induced antibody levels in infants better than T cells and is consistent with studies reporting baseline immunological signatures that are predictive of vaccine responses.⁴³”

Fig 1 is pretty superfluous as several groups have published on the immune trajectory post birth so this data is not novel. There have been groups looking at immune functions post SARSCoV-2 exposure which were not cited.

We do acknowledge that several studies have reported immune trajectory following birth, we included these results to demonstrate concordances with earlier studies and that differences reported here are associated with maternal HIV infection. Figure 1 is now included as the supplementary information in Figure S2B-F.

Sections: “Immune cell transition from birth to 9 months of age.” – **Line 114-139**, and “Distinct temporal maturation trajectories of NK cells stabilizing at 4 months coinciding with rapid CD8⁺ T cell divergence.”- **Line 141-187**.

We were unable to identify studies reporting SARS-Cov2 exposure associated to immune cellular development during infancy. Most studies report SARS-Cov2 infection during pregnancy described humoral immune changes occurring in their respective infants.

One of the main flaws is there was actually no significant differences in the NK cell groups between HEU and HUU (Suppl 2C) so really the reference to this in the text (line 229-234) should be removed. All of these points are not significantly different after adjustment-which is rightly used in these studies...and even when looking at the PCA there was only one significant

difference at week 4-when all other time points were equivalent. Hence, the data doesn't really support the conclusions 'NK cell phenotype and cluster composition differed between iHEU and iHUU at birth and week 4'-this wasn't significant at birth yet this is implied in this sentence.

The references to differences in NK cells have been modified to indicate that these differences were not significant following correction for multiple comparison.

Line 213-218:

“The composition of NK cells differed by HIV-exposure only at week 4 (Figure 2A), partly driven by higher frequencies of the differentiated (CD56^{lo}NKG2A⁻CD57^{lo/+})³⁰ NK cell cluster 1 (CD56^{lo}CD16^{lo} Perforin⁺CD38⁺CD45RA⁺FcER1⁺, p=0.04, p.adj=0.2) and cluster 5 (CD56^{lo}CD16⁺Perforin⁺CD57⁺CD45RA⁺CD38⁺ (p=0.03, p.adj=0.2) in iHEU compared to iHUU (Figure S4A), **although these difference were not significant after correcting for multiple testing.**”

We have also revised the conclusion stating that “NK cell phenotype differed between iHEU and iHEU at birth and week 4”.

Line 219-221:

“These findings show that HIV/ARV exposure sequentially disrupts immune trajectory over time after birth beginning with subtle changes in innate NK cells and later disrupting memory T cell maturation.”

Line 394-396:

“HIV exposure altered the immunological compositions of NK cells and T cells at different stages during infancy, with the T cell memory maturation being significantly impacted later in life.”

Although significant differences in clusters between HEU and HUU as a group were seen in CD4 T cells for example (Fig 2), the heat plots demonstrate many HEU samples (nearly half in some clusters) did still look similar to HUU-what is the difference between these different HEU samples? Unbiased clustering may put those samples into different groups, some more like HUU and some not? Are there any differences between the infants within the HEU that may explain this and that you could comment on?

We do observe a bimodal distribution among iHEU when accessing immunological profiles. To understand which factors are responsible for this outcome we assessed whether any of the demographic characteristics such as timing of maternal ARV initiation, CD4⁺ T cell count, and duration of breastfeeding could explain these outcomes, however we could not detect significant differences for any of these parameters.

Why do the heat plots for CD8 T cells in Fig 2 have less samples in them than CD4? They should be the same? Where is the rest of the data for CD8 from the samples that CD4 was measured in?

For our mass cytometry data pre-processing analysis, immune subsets (NK cells and CD4 with <1000 cells/events were excluded from downstream analysis such as the unsupervised cell clustering. CD4 cells were more abundant in the infant PBMC samples compared to CD8 cells and NK cells and therefore were more likely to be excluded. This additional preprocessing step is highlighted in the methods section.

Line 630-633:

“Post CyTOF acquisition, the files were normalized using Premessa R package and target populations were manually gated using FlowJo software (version 10.5.3, TreeStar). The target populations with >1000 events were exported as FCS files for downstream analysis using R, an open-source statistical software⁷⁴”

How many memory cells were sorted from the HEU/HUU from which the TCR repertoire was assessed-this could skew the data as the memory cell numbers are low?

Supplementary Table S3 has been included detailing median cell number for both naïve and memory T cells used for TCR sequencing. There was no significant difference observed between the memory cells between iHEU and iHUU.

Line 233-239:

“Of the remaining 861 samples included in the study analysis, there was a total of 238,092 reads (range: 11-5451), which differed between iHEU and iHUU for naïve CD4⁺ T cells at weeks 36 (Tables S3). The number of unique clones identified per T cell subset positively correlated with the total number of reads (Figure S3B). The median number of unique clones

per T cell subset, however, was higher in memory T cells for iHUU compared to those of iHEU at birth, and for naïve CD4⁺ T cell a similar difference was observed at week 36 (Table S3).”

Similarly, it is not clear how many individual samples were analysed for the TCR repertoire-you state 934 sorted cell fractions were assessed which is not divisible by 4 implying you didn't sort 4 fractions from every sample? This all needs to be clearer and more transparent.

A consort Figure S1 has been included showing number of study participants and number of samples analysed for mass cytometry, TCR-sequencing and antibody responses.

The TCR repertoire and naïve T cell cluster correlations that you make-were these done on the total cell cluster analysis or only using samples from which the TCR repertoire was also assessed? Were cell correlations with TCRb diversity only seen in HEU and not in HUU at birth? It isn't clear if that correlation was assessed and was not apparent?

The TCR diversity is derived from the naïve and memory T cells was correlated to the T cell cluster abundances for each matched sample as determined from the mass cytometry analysis. Figure 3C only show correlations that were statistically significant after correcting for multiple comparisons. We further added to the in-text reference to indicate that the correlations were done for all time points for samples that had both TCR and mass cytometry data. We have further clarified that the significant correlations were only apparent for iHEU.

Line 273-290:

“Spearman’s rank correlation between T cell clusters and the inverse Simpson TCRβ diversity scores for naïve and memory CD4⁺ and CD8⁺ T cells were measured for infants with paired TCRβ and mass cytometry data at all time points. Here, the TCRβ diversity scores derived from sorted naïve T cells in iHEU were positively correlated to the frequencies of CD4⁺ T cell clusters (4-6, 9-11) and CD8⁺ T cell clusters (2,3,7) (Figure 3C). It is noteworthy that the observed T cell clusters that were correlated to the naïve TCRβ diversity included the HLA-DR⁺Ki67⁺CD4⁺ T cell clusters (4 & 11), memory differentiated CD4⁺ T cell clusters (5,6, 9 & 11) and the CD8⁺ T cell clusters (2,3 & 7) that were significantly lower amongst iHEU at week 15 and 36 (Figure 2D-G). Relative to the naïve CD4⁺ and CD8⁺ T cell clusters, the T cell clusters that were correlated with TCRβ displayed varying expression levels of PD-1 suggesting TCR activation (Figure S3B & S3C)^{34,35}. For the proliferating activated CD4⁺ T cells (cluster 4 & 11) and the terminally differentiated cytolytic CD4⁺ T cells (cluster 5 & 6), these correlations were

evident from birth up until week 15. Although a similar trend was observed for the T cells clusters and TCR diversity derived from sorted memory T cells, the observed correlation was not statistically significant after correcting for multiple correction (data not shown)."

Whilst we may assume that the infants in this study were born by vaginal delivery and not caesarean section, this is not explicitly stated and as delivery mode is a key variable in ongoing immune function this needs to be clarified. Also, there was a significant difference in age of mothers between the two groups but this was not discussed.

All infants included in the study analysis were born by vaginal delivery. This has been indicated in the methods.

Line 515: "...enrolment was restricted to infants who were born by vaginal delivery..."

There is a lot of data regarding the role of Tregs post birth but no Tregs identified in the analysis? It seems that FOXP3 was not in the Ab panel, this needs to be discussed as a limitation.

We previously reported on the longitudinal changes in Treg cells (ref #17) in a similar cohort of iHEU and iHUU. This work formed part of the premise for the mass cytometry study.

Line 76-80:

"We have previously shown that maternal HIV/ARV exposure alters the dynamics of the T regulatory (Treg) to Th17 cell ratio resulting in a Th17/Treg imbalance associated with gut damage¹⁷. In this paper, we extend this analysis to investigate the impact of HIV/ARV exposure on T cell clonality, memory and NK cell maturation differences between iHEU and iHUU."

There were a lot of typos/missing information. Please correct.

Thank you for bringing this to our attention, we have thoroughly reviewed and revised the manuscript to correct all typographical and grammatical errors.

INF γ lines 86/87/88. Be consistent IFN-g or INF-g

For consistency, interferon gamma has been abbreviated as IFN- γ throughout the manuscript.

Line 157 parsed manually gated CD4 and CD8 cells (presumably paired?)

The CD4 and CD8 cells were paired as they were derived from the CD3⁺ T cell population. However, to avoid misclassification of cell cluster due to few cells, cell subsets (CD4 or CD8 T cells) with fewer than 1000 cells were removed from further analysis. This has been reflected in the methods.

Line 144-146: "...manually gated on CD4⁺ and CD8⁺ T cells from the CD3⁺ T cell population and targeted the NK cell population using lineage-exclusion manual gating (Figure S2A)."

Line 630-633:

"Post CyTOF acquisition, the files were normalized using Premessa R package and target populations were manually gated using FlowJo software (version 10.5.3, TreeStar). The target populations with >1000 events were exported as FCS files for downstream analysis using R, an open-source statistical software."

Unclear why 'expanded NK cell subsets in iHEU' would be detrimental and not a positive- please expand or remove.

Statement relating to expanded NK cells as detrimental has been removed.

Not sure Per is a well known abbreviation for perforin-change and also be consistent, sometimes this is labelled as Per, other times perforin.

To avoid confusion all instances where perforin was abbreviate as "Per" has been replaced with "perforin".

Table S1/S2 are in the wrong order as S2 discussed first in the manuscript?

We have corrected the order of supplementary tables to reflect the order of appearance in the manuscript; Table S1 Infant cohort characteristics and Table S1 List of key resources.

Please number the clusters in Fig S2 so we can see how they relate to the clusters in Fig 2-not just colour coded.

Fig. 2 and Fig. S2 have been consolidated to Fig 1 and Figures showing cell clusters are annotated by colour and numbering.

Cluster 5 in CD8, what other markers are on this cluster -this is distinguished from cluster 1 by presence of CD45RA, CD27?'

Table 2 describing the phenotypes of the CD8⁺ T cell clusters has been revised to highlight the markers expressed by each individual cluster. This information is also included in the heatmap showing median expression of the cell markers for each cluster (**Figure S3c**). The phenotype for CD8⁺ T cell cluster 5 are cells expressing Perforin⁺CD57⁺CD45RA⁺**DNAM1⁺LILRB1⁺2B4⁺** and distinguishable from cluster 1 that express Perforin⁺ CD57⁺CD45RA⁺**CD38⁺HLA-DR⁺CCR4⁺Siglec-7⁺**.

X axis labels on Fig 4a are absent?

X-axis for Figure 4a (now **Figure 3a**) have been included showing the different age visit for the infants with which the samples for TCR-seq were collected.

Fig 4c/d p=0.05/p=0.02 have same size circles in the key. remove p=0.75 info from the dataset as ns and I have no clue what the tiny circles represent as no key for those at all!

Figure 4c/d [now **Figure 3c**] has been revised to correctly annotate the adjusted p-value for the Spearman's correlation analysis for each cluster with the TCR data.

Did the authors assess any other populations to see if they related to subsequent Ab responses -B cells, APCs? Did you only put the T/Nk data into the prediction model?!

The mass cytometry panel was designed to exhaustively characterise NK cells and T cells, and only lineage markers were included for the exclusion of other cell populations such as B cells and monocytes. This therefore limited our antibody predictive model to the NK and T cell clusters detected in the infant samples.

Whilst the authors suggested a functional link between NK cell and Ab responses, more discussion about the ACTUAL NK populations that were linked compared to others that were not-what do those specific populations suggest? If you did the prediction model between responders and non-responders but further separated based on HEU/HUU what happens? Do any of the correlations change if you separate based on HIV exposure?

Figure 5 [now **Figure 4**] has been revised to also show how the NK cells relate to antibody responses and we have included a discussion to highlight the implication of these findings.

Results

Line 360-365:

“NK cell cluster 1 (CD56^{lo}CD16^{lo} Perforin⁺CD38⁺CD45RA⁺FcεRIγ⁺) and cluster 3 (CD56⁻CD16⁺FcεRIγ⁺) measured at birth and week 4 were consistently associated with responders to pertussis vaccine at week 15, while cluster 8 (CD56⁺CD16⁺Perforin⁺CD38⁺CD45RA⁺DNAM1⁺2B4⁺Nkp30⁺NKp46⁺NKG2A⁺Siglec-7⁺FcεRIγ⁺) was associated with non-responders to pertussis vaccine (Figure 5E).”

Line 379-381:

“Similar to the anti-pertussis response, NK cell cluster 1 was also associated with high anti-rotavirus IgA responses, while NK cell cluster 8 was associated with low anti-rotavirus IgA (Figure 5G).”

Discussion

Line 456-481:

“To better understand how these alterations in iHEU immunity could impact immunological responses we measured antibody response to common childhood vaccines such as to pertussis and rotavirus. Previous studies have consistently demonstrated that although maternally derived vaccine specific antibodies in iHEU are lower at birth, following vaccination the concentrations tend to increase to similar levels to those of iHUU, and for other vaccines, such as acellular pertussis vaccine and pneumococcal conjugate vaccine (PCV), where iHEU antibody levels even surpass that of iHUU, albeit with poor functionality.⁶⁴ In our cohort we also observed a similar trend for anti-pertussis IgG levels being higher in iHEU compared to iHUU. Our data further revealed that among iHEU, vaccine-specific responses tend to have a high degree of variance compared to iHUU. This enabled us to classify infants either as “poor” or “good” vaccine responders, and therefore allowing us to show that, instead of T cells, the composition of NK cells pre-vaccination was superior in predicting vaccine-induced antibody responses^{2330,6521} postvaccination. This was consistent with other studies showing innate immune cells, including NK cells, being good baseline markers for predicting vaccine-induced antibody responses, albeit in adults.^{43,66} NK cells can either negatively or positively regulate adaptive immunity, including promoting immunoglobulin isotype switching and enhancing antibody production^{67,68}. Here we show that the abundance of

CD56^{lo}CD16^{lo}CD38⁺CD45RA⁺FcεR1γ⁺ NK cell subset prior to immunization resulted in a “good” vaccine-induced antibody response. This may be in agreement with Cuapio et al., who identified a CD56^{lo}NKG2C⁺ NK subset whose frequency at baseline correlated with anti-SARS-Cov2 antibody titres post-vaccination.⁶⁶ These data suggest: a) that NK cell immunity may be important for vaccine-induced adaptive immunity and b) induction of NK cells soon after birth could be an approach to enhance vaccine immunogenicity for childhood vaccines. Further mechanistic investigations are required to validate these observations.”

Why are responses to one vaccination greater in HEU than HUU but vice versa for a different vaccination? Some discussion should be made of this.

The differences for the rotavirus antibody response were not statistically significant. However, there are several studies that have reported inconsistent outcomes related to vaccination of iHEU, these have been discussed in the revised version of the manuscript.

Line 458-463:

“Previous studies have consistently demonstrated that although maternally derived vaccine specific antibodies in iHEU are lower at birth, following vaccination the concentrations tend to increase to similar levels to those of iHUU, and for other vaccines, such as acellular pertussis vaccine and pneumococcal conjugate vaccine (PCV), where iHEU antibody levels even surpass that of iHUU, albeit with poor functionality.⁶⁴”

Reviewer #3

The implementation improved and safe antiretroviral therapy options has resulted in a drastic reduction of in utero infection of infants with HIV. However, the concurrent increase in HIV-exposed uninfected infants presents as another serious health problem, as many iHEU experience increased susceptibility to infections. The study by Dzanibe et al. aims to identify underlying immune differences between iHEU and HIV unexposed and uninfected infants. Performing mass spectrometry, the investigators identified defined the immune trajectory of CD4⁺ and CD8⁺ T cells, as well as NK cells in the first 9 months of life. The results demonstrated differences in the differentiation of specific T cell clusters between iHEU and iHUU, especially at early time points. In addition, memory T cell populations of iHEU exhibited reduced TCR beta clonotypic diversity compared to iHUU. The immune analysis further

revealed divergence between NK cell populations depending on the infant's HIV exposure. Independent of HIV exposure, the presence of specific NK cell populations at birth was associated with vaccine responses to rotavirus vaccine and acellular pertussis vaccine at 9 months of age.

Together, the presented findings add to our knowledge of immune development in African infant populations and provide new insights into kinetic and qualitative differences of immune ontogeny between iHEU and iHUU. As such, the study results are significant.

The clarity of the manuscript could be improved, and the following comments should be addressed:

1. The authors talk about a longitudinal study and in the Discussion (lines 461ff) they refer to the "...novelty of our study lies with the longitudinal follow-up of our infants with matched samples...". Please clarify, if this statement applies only to the infant samples from weeks 4, 15, and 36 (n=53, n=52, and n=53, respectively). Do the samples of cord blood (n=5) and 4-week-old infants (n=44) represent different groups of infants?
2. How were matched samples versus cross-sectional samples considered in the various analyses, and in particular in the immune trajectory analysis? Considering the high variability in data at each time point, would a separate analysis using only matched samples provide more concise data?
3. The immune panel was clearly biased towards T cell and NK cell populations. Monocyte, dendritic cell, and B cell differentiation was not tested. It might be possible to include monocytes with characterization by CD14 and CD16, this might be interesting for iHEU? This should be stated in the methods. Please also provide a rationale.
4. Considering the association between NK cell populations at birth and vaccine responses at wk 36, one wonders if there would have been correlations between antigen-presenting cell populations and vaccine responses, too, and whether B cell clusters and B cell differentiation would correlate with vaccine responses. The lack of any associations between T cell populations and vaccine responses could also be explained by the lack of consideration of follicular T helper cell responses in the current study.
5. The manuscript title implies that this study was focused on differences between iHEU and iHUU in T cell development. Yet, even in the abstract, the authors point out the differences in NK cell development and point to the important finding that birth NK cell clusters correlate with vaccine responses. The latter correlation, however, is independent of HIV exposure. Therefore, the reviewer wonders whether different sets of interesting data were combined in this manuscript. The link between the data is not always clear and sometimes almost artificial.

Maybe, data of general immune development for T cells and NK cells should proceed the description of developmental differences between iHEU and iHUU? For lack of functional studies, one could then envision examining TCR diversity and vaccine-induced antibody responses.

6. The finding that iHUU have a preference for HIV among the TCR beta clones is curious and worth some discussion.

7. On Page 10, line 229, should “iHEU” read “iHUU had higher frequencies...”?

8. Please provide a reference for your statement that immune populations at week 36 had an adult-like phenotype (page 7, line 151).

Rebuttal:

1. *The authors talk about a longitudinal study and in the Discussion (lines 461ff) the refer to the “..novelty of our study lies with the longitudinal follow-up of our infants with matched samples...”. Please clarify, if this statement applies only to the infant samples from weeks 4, 15, and 36 (n=53, n=52, and n=53, respectively). Do the samples of cord blood (n=5) and 4-week-old infants (n=44) represent different groups of infants?*

All samples included in the study analysis were matched across all time points. Although only 5 infants had matching cord blood samples available for our study analysis, we decided to include these CBMC samples to determine how cord blood differs to infant peripheral blood collected at birth (<12hr).

Furthermore, due to varying cell counts and cell events from the cytometer, sample with too few cells (<1000 events) were excluded from further analysis. Since only small blood volumes could be collected from the earlier visit (especially at birth), these samples tend to have had fewer samples and were more likely to be excluded in downstream analysis. We have included a consort diagram (Figure S1) to the number of samples included for each visit for all the end-point analysis performed for this study. This additional preprocessing step is highlighted in the methods section.

Line 630-633:

“Post CyTOF acquisition, the files were normalized using Premessa R package and target populations were manually gated using FlowJo software (version 10.5.3, TreeStar). The target populations with >1000 events were exported as FCS files for downstream analysis using R...”

- 2. How were matched samples versus cross-sectional samples considered in the various analyses, and in particular in the immune trajectory analysis? Considering the high variability in data at each time point, would a separate analysis using only matched samples provide more concise data?*

Since only a few participants did not have all samples at varying study visits, we opted to include all samples in the various analysis as described in Figure S1. Figures 1A-F showing immune trajectory of NK cells and T cells were revised to demonstrate which of the samples were matched across the different study visit and how this reflected an individual's immune trajectory. The prediction models were only restricted to infants with matching samples for both the early time points (birth and week 4) and the later time points (weeks 15 and 36).

- 3. The immune panel was clearly biased towards T cell and NK cell populations. Monocyte, dendritic cell, and B cell differentiation was not tested. It might be possible to include monocytes with characterization by CD14 and CD16, this might be interesting for iHEU? This should be stated in the methods. Please also provide a rationale.*

See response to point 4.

- 4. Considering the association between NK cell populations at birth and vaccine responses at wk 36, one wonders if there would have been correlations between antigen-presenting cell populations and vaccine responses, too, and whether B cell clusters and B cell differentiation would correlate with vaccine responses. The lack of any associations between T cell populations and vaccine responses could also be explained by the lack of consideration of follicular T helper cell responses in the current study.*

This point is well taken. However, the primary focus of the study was to assess the impact of maternal HIV infection on NK cells and T cells. Thus, the mass cytometry panel was designed to characterise these phenotypes longitudinally. We do acknowledge that post hoc analysis of the predictive potential of other immune cells specifically those involved in antigen presentation (monocytes), antibody production (B cells) and regulation (follicular T helper cells) could provide better insight on the measured antibody responses. These findings show for the first time that baseline immune cells in infant are predictive of vaccine-induced antibody titres later in life and thus warrants further investigation.

- 5. The manuscript title implies that this study was focused on differences between iHEU and iHUU in T cell development. Yet, even in the abstract, the authors point out the*

differences in NK cell development and point to the important finding that birth NK cell clusters correlate with vaccine responses. The latter correlation, however, is independent of HIV exposure. Therefore, the reviewer wonders whether different sets of interesting data were combined in this manuscript.

The data sets presented in the manuscript were performed on matched infant samples, this included performing mass cytometry and TCR-seq analysis using infant PBMC samples and antibody levels measured on paired plasma samples. Since there were fewer samples having both mass cytometry data and antibody measurements at all time points, we could not stratify this analysis by HIV exposure. We opted to determine if the identified cluster could provide preliminary insight of how these would influence vaccine responses as a proxy of functional outcomes in our study. We have revised the manuscript to reflect the main findings of the study were the altered T cell memory differentiation that is preceded by skewed TCR clonality.

Abstract Line 35-41:

“Using mass cytometry, we show alterations in T cell memory differentiation between iHEU and iHUU being significant from week 15 of life. The altered memory T cell differentiation in iHEU was preceded by lower TCR V β clonotypic diversity and linked to TCR clonal depletion within the naïve T cell compartment. Compared to iHUU, iHEU had elevated CD56^{lo}CD16^{lo}Perforin⁺CD38⁺CD45RA⁺Fc ϵ R1y⁺ NK cells at 1 month postpartum and whose abundance pre-vaccination were predictive of vaccine-induced pertussis and rotavirus antibody responses post 3 months of life.”

The link between the data is not always clear and sometimes almost artificial. Maybe, data of general immune development for T cells and NK cells should proceed the description of developmental differences between iHEU and iHUU? For lack of functional studies, one could then envision examining TCR diversity and vaccine-induced antibody responses.

We have restructured the results to first describe the longitudinal development of the overall infant immune system. This approach reveals stereotypic lineage immune trajectory that corroborates earlier studies and thus validating our data analysis (**Figure S2**). We then proceeded to show how *in utero* exposure to HIV alters the composition of T cells of iHEU to diverge from the immune maturational trajectory relative to iHUU later in life (**Figure 2B & C**). We further describe which specific T cell populations were responsible for the varying T cell compositions in iHEU by highlighting that these included similar T cell clusters based on hierarchical clustering (**Figure S3B & C**) and that these clusters were lower in iHEU compared

to iHUU (generalised linear modelling testing: **Figure 2D-G**). The differences in lower memory differentiated T cells coincided with lower V β TCR diversity specifically for the naïve populations of CD4⁺ and CD8⁺ T cells (**Figure 3A**), which was further observed to be positively correlated with the T cell populations that differed by infant HIV exposure status (**Figure 3C**).

To clarify these findings the manuscript was revised as follows:

Results

Line 193-204 –“PCA components derived from the centred log-ratios of the relative cluster abundances for each cell subsets were compared between iHEU and iHUU to determine compositional differences. There were statistically significant differences at weeks 15 and 36 for CD4⁺ T cells and at week 36 for CD8⁺ T cells (Figure 2B & 2C). Differential abundance testing using generalized mixed model revealed that the divergent CD4⁺ T cell phenotypes in iHEU were due to lower frequencies of closely related activated and proliferating (HLA-DR⁺Ki67⁺) clusters 4 & 11, and memory differentiated cytolytic (PD-1⁺CD57⁺Perforin⁺)^{31,32} clusters 5- 9 compared to iHUU at week 15, with the differentiated cytolytic clusters 5, 7 and 9 remaining significantly lower in iHEU at week 36 (Figure 2D, 2E & S3B). The CD4⁺ Th2-like cluster 1 (CCR4⁺CD27⁺CD127⁺) was, however, significantly higher in iHEU compared to iHUU at week 15 although by week 36 was no longer significantly elevated.”

Line 205-211 – “For CD8⁺ T cells, the compositional differences were partly driven by lower mean frequencies for clusters 2 (NKT-like cells: CD56⁺CD16⁺NKp30⁺NKp46⁺NKG2A⁺Perforin⁺)³³, 3 (Naïve-like activated cell: CD45RA⁺CD27⁺CCR7⁺CXCR3⁺CCR4⁺HLA-DR⁺) and 7 (proliferating EM: Ki67⁺ CD45RA⁺CD27⁺CCR7⁺Perforin⁺) in iHEU compared to iHUU at week 15 (Figure 2I). By week 36, only the mean frequencies of clusters 2 (NKT-like cells) and 3 (Naïve-like cells) remained significantly lower in iHEU compared to iHUU (Figure 2F & 2G).”

Line 253-256 – “In contrast to TCR β of memory T cells, in which significant differences were found early life, within the naïve T cell compartment; TCR β diversity was significantly lower in iHEU compared to iHUU at week 15 and 36 for CD4⁺ T cells, and at birth and week 36 for CD8⁺ T cells (Figure 3A).”

Line 270-273 – “Since the naïve TCR β clonotypic differences coincided with the lower memory differentiated T cell clusters (Figure 2), we next wished to determine if there was a relationship between TCR β diversity and the clusters of CD4⁺ and CD8⁺ T cells identified.”

Line 276-282 - “the TCR β diversity scores derived from sorted naïve T cells in iHEU were positively correlated to the frequencies of CD4⁺ T cell clusters (4-6, 9-11) and CD8⁺ T cell clusters (2,3,7) (Figure 3C). It is noteworthy that the observed T cell clusters that were correlated to the naïve TCR β diversity included the HLA-DR⁺Ki67⁺CD4⁺ T cell clusters (4 & 11), memory differentiated CD4⁺ T cell clusters (5,6, 9 & 11) and the CD8⁺ T cell clusters (2,3 & 7) that were significantly lower amongst iHEU at week 15 and 36 (Figure 2D-G).”

Discussion

Line 394-403:

“HIV exposure altered the immunological compositions of NK cells and T cells at different stages during infancy, with the T cell memory maturation being significantly impacted later in life. The impaired T cell memory maturation among iHEU could be in part be due to diminished naïve TCR clonal pools observed in iHEU relative to iHUU. The skewing of TCR clonality most likely initiating *in utero* as suggest by difference in infant groups observed as early as birth.^{41,42} We further show that pre-vaccine NK cell composition can predict vaccine-induced antibody levels in infants better than T cells and is consistent with studies reporting baseline immunological signatures that are predictive of vaccine responses.⁴³”

6. *The finding that iHUU have a preference for HIV among the TCR beta clones is curious and worth some discussion.*

We have added an extensive discussion to the manuscript to explain the possible reasons for the observed outcome.

Line 424-455:

“Gabriel *et al.* reported lower TCR β clonality in the cord blood of iHEU,⁴¹ implying clonal expansion occurring *in utero*. Our data further revealed that memory T cells were the main contributors of reduced TCR β diversity among iHEU compared to iHUU at birth. In addition to lower TCR diversity at birth, iHEU also had altered V β gene usage, with several genes being used more frequently relative to iHiUU. When using GLIPH2^{36,37}, however, we could not detect any predicted antigen specificity enriched among iHEU including HIV-specific clones. This contrasted with earlier studies showing higher frequencies of HIV-1 specific clones⁴¹ and reactivity of T cells towards HIV proteins in iHEU⁵⁶. What was evident instead was that naïve

CD4⁺ and CD8⁺ TCR β specificities in iHUU were enriched for clones targeting CMV, EBV and HCV antigens.

Since we obtained >95% purity after sorting for naïve and memory T cells, we regard this as minimal contamination of naïve T cells by memory T cells that could account for these enriched clones. It is possible that the abundance of naïve T cells compared to memory T cells in infant PBMC contributed to sampling bias resulting in enriched TCR clones amongst naïve T cells compared to the memory fraction. We also rationalized that since we did not include CD95 in our panel, we could not tease out stem cell-like memory T cells that would also express CD45RA and CCR7 similar to naïve cells and thus contribute to the observed outcome^{57–59}.

These scenarios, however, do not explain the lack of antigen specific TCR clonal enrichment among iHEU who exhibited skewed TCR clonality and V β gene usage. In addition, the length of CDR3 derived from these clones were typically short, a phenomenon common with neonatal TCR repertoire, and did not differ between iHEU and iHUU. Altogether, these observed outcomes suggest other non-antigen driven mechanisms that resulted in the loss of diversification of the iHEU TCR repertoire. Factors such as homeostatic proliferation under chronic maternal immune activation and proinflammatory milieu could give rise to virtual T cells,^{60–62} altered thymic selection since iHEU have been reported to have reduced thymic size relative to iHUU,^{42,63} or partially dysfunctional RAG system could contribute to iHEU having skewed TCR clonality. Since it is evident that reduced TCR diversity in iHEU is associated with impaired T cell memory differentiation and possibly increasing vulnerability to infection among iHEU,⁵³ further investigations are required to decipher the mechanism responsible for the skewed TCR clonality in iHEU.”

7. On Page 10, line 229, should “iHEU” read “iHUU had higher frequencies...”?

Line 229 (now **Line 213-217**) correctly state that “composition of NK cells differed by HIV-exposure only at week 4 (Figure 2A), partly driven by higher frequencies of the differentiated (CD56^{lo}NKG2A⁻CD57^{lo/+})³⁰ NK cell cluster 1 (CD56^{lo}CD16^{lo} Perforin⁺CD38⁺CD45RA⁺FcER1y⁺, p=0.04, p.adj=0.2) and cluster 5 (CD56^{lo}CD16⁺Perforin⁺CD57⁺CD45RA⁺CD38⁺ (p=0.03, p.adj=0.2) in iHEU compared to iHUU (Figure S4A)”

8. Please provide a reference for your statement that immune populations at week 36 had an adult-like phenotype (page 7, line 151).

Reference #30 has been included to support the statement **Line 184-187** “these temporal phenotypic changes in cell clustering illustrate the archetypal progression and transition of infant immunity from neonatal to an adult-like immune phenotype by 9 months of age, being characterised by a shift from innate myeloid cells to an increase in lymphoid adaptive immune cells³⁰.”

REVIEWERS' COMMENTS

Reviewer #1 (Remarks to the Author):

The authors have addressed all comments/questions from the initial review. Accept

Reviewer #2 (Remarks to the Author):

The manuscript is greatly improved and most of my comments have been acted upon. Even though the authors failed to identify studies reporting SARS-Cov2 exposure associated to infant immune cellular development, there are many papers and reviews looking at SARS Cov-2 exposure in mothers and immune cell function in infants and at least one should be cited. eg J Clin Med 2023 Jun 25;12(13):4256. doi: 10.3390/jcm12134256.

Front Pediatr 2022 Nov 7;10:1046100.doi:10.3389/fped.2022.1046100.

Pediatr Res 2022 Apr;91(5):1090-1098. doi: 10.1038/s41390-021-01793-z. Epub 2021 Nov 8.

Nat Immunol. 2021 Dec;22(12):1490-1502.doi: 10.1038/s41590-021-01049-2. Epub 2021 Oct 6.

Also, the significant difference in age of mothers between the two groups was still not commented on...

Reviewer #3 (Remarks to the Author):

The authors have been very responsive to the critiques. In my opinion, the result are presented in a more logical and easier to understand order.

While there remains variability in data, the reviewer acknowledges the difficulties of obtaining longitudinal infant samples and working with small blood volumes. The presented analyses are comprehensive and the data enrich the still minimal data sources we have on immune ontogeny using longitudinal samples from African infants. AS there will be an increasing number of HEU infants in the years to come, there is a need to understand differences in immune ontogeny between HEU and HUU to identify critical factors that increase morbidity in HEU infants.

Editor: Nature communication

RE: Manuscript revision: Delayed memory T cell expansion in HIV-exposed uninfected infants is preceded by premature skewing of T cell receptor clonality. (NCOMMS-23-23072A-Z)

We thank the reviewers for their constructive feedback in improving our manuscript. Herein, we describe the revision for the above-referenced manuscript. This rebuttal only addresses comments raised by reviewer 2, since they had additional recommendations.

Reviewer #2

The manuscript is greatly improved and most of my comments have been acted upon. Even though the authors failed to identify studies reporting SARS-Cov2 exposure associated to infant immune cellular development, there are many papers and reviews looking at SARS Cov-2 exposure in mothers and immune cell function in infants and at least one should be cited.

eg J Clin Med 2023 Jun 25;12(13):4256. doi: 10.3390/jcm12134256. Front Pediatr 2022 Nov 7;10:1046100.doi:10.3389/fped.2022.1046100.
Pediatr Res 2022 Apr;91(5):1090-1098. doi: 10.1038/s41390-021-01793-z. Epub 2021 Nov 8.
Nat Immunol. 2021 Dec;22(12):1490-1502.doi: 10.1038/s41590-021-01049-2. Epub 2021 Oct 6. Also, the significant difference in age of mothers between the two groups was still not commented on...

Even though the authors failed to identify studies reporting SARS-Cov2 exposure associated to infant immune cellular development, there are many papers and reviews looking at SARS Cov-2 exposure in mothers and immune cell function in infants and at least one should be cited.

Response: Thank you for these suggestions. We have included in our discussion how SARS-Cov2 influences newborn immune cells and how this compares to HIV. We have further extended our discussion to include exposure to other viruses such as CMV. We have inserted the following and also highlighting how important it is to do longitudinal studies:

Extract **Line 428-434**

“Other viral infections occurring during pregnancy such as SARS-Cov2 have also been documented to alter infant immunity even in the absence of viral transmission^{53,54}. Although these studies were limited in examining the difference in cord blood, early viral exposure

resulted in lower frequencies of TNF- α and IFN- γ producing T cells⁵³. Further longitudinal investigations are required to determine the impact of these early life cellular immune changes and their association to clinical outcomes later in life.”

Extract **Line 443-445**

“Gabriel *et al.* reported lower TCR β clonality in the cord blood of iHEU⁴² implying clonal expansion occurring *in utero*, a phenomenon also observed in SARS-Cov2 exposed to infants⁵⁴”

Also, the significant difference in age of mothers between the two groups was still not commented on...

Response: In our cohort the overall median age of mothers living with HIV was significantly higher compared to those living without HIV. We have included in the manuscript a comment regarding this difference in the population characteristics.

Extract **Line 120-122**

“Our infant cohort consisted of 36 infants (iHEU=40 and iHUU=16) and having similar population characteristics include birth weight and gestational age, with the exception that the mothers of iHEU were noted to be significantly older (Table S1).”

Sincerely,

Clive Gray MSc, PhD, ASSAf
Professor, Division of Immunology, Biomedical Research Institute